# Realisation of de Gennes' absolute superconducting switch with a heavy metal interface

Hisakazu Matsuki[1], Alberto Hijano[2,3,4], Grzegorz P. Mazur [1,5], Stefan Ilić[2,4], Binbin Wang[6], Iuliia Alekhina[1], Kohei Ohnishi [7], Sachio Komori [8], Yang Li [1,9], Nadia Stelmashenko [1], Niladri Banerjee [10], Lesley F. Cohen [10], David W. McComb [6], F. Sebastián Bergeret [2,11], Guang Yang [1,12,13] ✉ & Jason W. A. Robinson [1] ✉

In 1966, Pierre-Gilles de Gennes proposed a non-volatile mechanism for switching superconductivity on and off in a magnetic device. This involved a superconductor (S) sandwiched between ferromagnetic (F) insulators in which the net magnetic exchange field could be controlled through the magnetisation-orientation of the F layers. Because superconducting switches are attractive for a range of applications, extensive studies have been carried out on F/S/F structures. Although these have demonstrated a sensitivity of the superconducting critical temperature ($T_c$) to parallel (P) and antiparallel (AP) magnetisation-orientations of the F layers, corresponding shifts in $T_c$ (i.e. $\Delta T_c = T_{c,AP} - T_{c,P}$) are lower than predicted with $\Delta T_c$ only a small fraction of $T_{c,AP}$, precluding the development of applications. Here, we report EuS/Au/Nb/EuS structures where EuS is an insulating ferromagnet, Nb is a superconductor and Au is a heavy metal. For P magnetisations, the superconducting state in this structure is quenched down to the lowest measured temperature of 20 mK meaning that $\Delta T_c/T_{c,AP}$ is practically 1. The key to this so-called 'absolute switching' effect is a sizable spin-mixing conductance at the EuS/Au interface which ensures a robust magnetic proximity effect, unlocking the potential of F/S/F switches for low power electronics.

The original superconducting switch[1] modelled by de Gennes requires a thin-film superconductor (S) with a thickness ($d_s$) that is less than one superconducting coherence length ($\xi_s$), sandwiched between two ferromagnetic (F) insulators (Fig. 1a, b). Due to the strong pair-breaking interaction between the S and F materials, the critical temperature ($T_c$) of the F/S/F structure is suppressed for a parallel (P) alignment of the magnetisation of the F layers. Conversely, if the magnetisation of the F layers aligns antiparallel (AP), the influence of the two F layers on the superconductivity cancels, in principle, meaning that the suppression of $T_c$ is reduced[1,2]. An equivalent

[1]Department of Materials Science & Metallurgy, University of Cambridge, Cambridge, UK. [2]Centro de Física de Materiales (CFM-MPC) Centro Mixto CSIC-UPV/EHU, Donostia-San Sebastián, Spain. [3]Department of Physics, University of the Basque Country UPV/EHU, Bilbao, Spain. [4]Department of Physics and Nanoscience Center, University of Jyväskylä, Jyväskylä, Finland. [5]QuTech and Kavli Institute of NanoScience, Delft University of Technology, Delft, The Netherlands. [6]Department of Materials Science and Engineering, The Ohio State University, Columbus, OH, USA. [7]Department of Electrical, Electronic and Communication Engineering, Kindai University, Osaka, Japan. [8]Department of Physics, Nagoya University, Nagoya, Japan. [9]Cambridge Graphene Centre, University of Cambridge, Cambridge, UK. [10]Department of Physics, Blackett Laboratory, Imperial College London, London, UK. [11]Donostia International Physics Center (DIPC), Donostia–San Sebastián, Spain. [12]National Key Laboratory of Spintronics, Hangzhou International Innovation Institute, Beihang University, Hangzhou, China. [13]School of Integrated Circuit Science and Engineering, Beihang University, Beijing, China. ✉e-mail: gy251@buaa.edu.cn; jjr33@cam.ac.uk

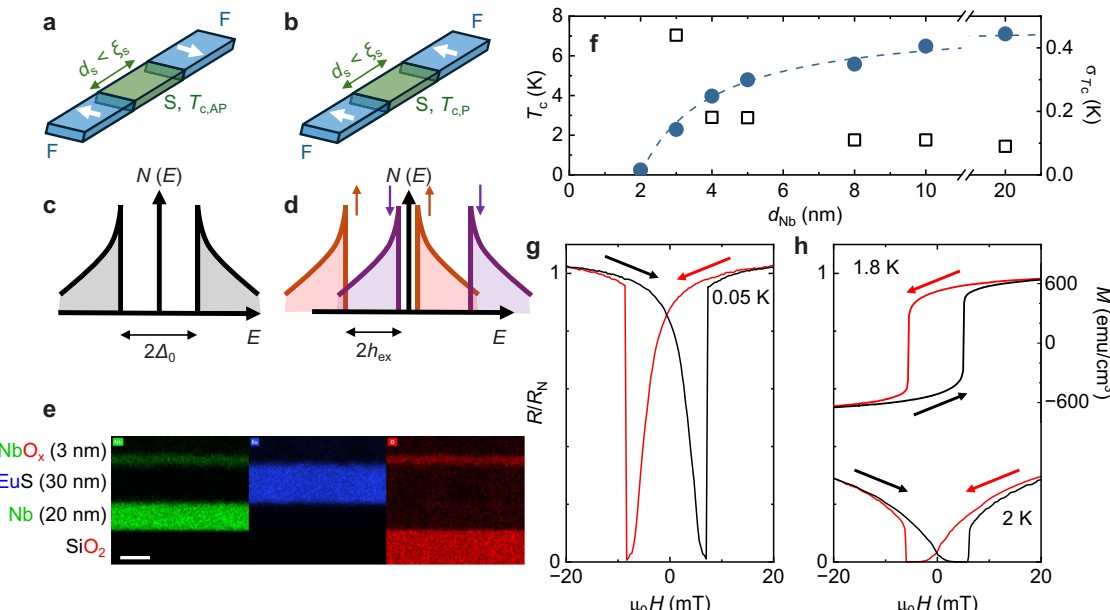

**Fig. 1 | A de Gennes' superconducting switch and structural, superconducting and magnetic properties of NbO$_x$(3 nm)/EuS(30 nm)/Nb($d_{Nb}$)/SiO$_2$//Si structures.** Schematic diagrams of a F/S/F superconducting switch in which a superconductor (S) is sandwiched between ferromagnetic insulators (F): **a** The proximity-induced magnetic exchange field ($h_{ex}$) in the S layer from the AP-aligned magnetisations is minimised or is, ideally, zero, preserving the superconducting state with a transition temperature $T_{c,AP}$; **b** For P-aligned magnetisations, $h_{ex}$ is maximised so the superconducting transition temperature $T_{c,P}$ is much lower than $T_{c,AP}$. **c, d** Representations of the superconducting density of states diagrams for the S layer for AP and P magnetisations of the F layers: **c** In the AP-state the density of states shows no evidence of proximity-induced magnetism (i.e. $h_{ex} = 0$), whereas in the P-state in (**b**) there is an energy splitting of $2h_{ex}$ in the spin-bands due to the proximity-induced exchange field. **e** STEM image from a control sample of a NbO$_x$(3 nm)/EuS(30 nm)/Nb(20 nm)/SiO$_2$//Si structure, showing the chemistry diagram with Nb (green), Eu (blue) and O (red). The scale bar has a length corresponding to 20 nm. **f** The left axis shows the zero-field-cooled superconducting transition temperature $T_c$ versus Nb thickness $d_{Nb}$ (blue) and the right axis shows the superconducting transition width $\sigma_{Tc}$ versus $d_{Nb}$ (black). **g** Normalised resistance $R$ versus in-plane magnetic field $H$ ($R(H)$) of an unpatterned NbO$_x$(3 nm)/EuS(30 nm)/Nb(2 nm)/SiO$_2$//Si structure at 50 mK, where $R_N$ is the normal state resistance. **h** Normalised $R(H)$ of an unpatterned NbO$_x$(3 nm)/EuS(30 nm)/Nb(3 nm)/SiO$_2$//Si structure at 2 K along with the magnetisation versus in-plane magnetic field $M(H)$ hysteresis loop for a 30-nm-thick EuS film at 1.8 K. Red (black) curves indicate a decreasing (increasing) in-plane magnetic field.

superconducting switch was later proposed by Tagirov[3], which involved transition metal ferromagnets (instead of F insulators), allowing superconductivity to penetrate the F layers, causing an additional background suppression of $T_c$ in both the P and AP magnetic states. Both models predicted that for certain parameter combinations, not only should the $T_c$ difference between P and AP magnetic states [i.e. $\Delta T_c = T_{c,AP} - T_{c,P}$] be a significant fraction of $T_{c,AP}$, but also that superconductivity should be completely suppressed for all temperatures in the P-state—this is so-called 'absolute switching' with $\Delta T_c/T_{c,AP} = 1$ meaning that F/S/F becomes a truly magnetically-controlled superconducting switch[4–9], a highly sought-after device for low power electronics.

Probably the first[10] experimental demonstration of F/S/F switching was reported in 2002 with measured values of $\Delta T_c$ (roughly 6 mK) much lower than predicted[1–3]. In addition, because the temperature width of the superconducting transition $\sigma_{Tc}$ was larger than $\Delta T_c$, the resistance change at any temperature induced by the magnetic reorientation was small. Since then many papers on F/S/F or similar switches have been published[11–23] using different materials combinations, largely involving transition metal ferromagnets in which the magnetic exchange field is dominated by spin-splitting of the $d$-orbitals and transport through hybridised $s$-$d$ orbitals; however, values of $\Delta T_c$ are generally much lower than predicted by theory. Well-defined on and off switching of superconductivity has been demonstrated in limited F/S/F structures involving $f$-orbital magnets such as metallic Ho[23,24] or insulators including EuS[21] and GdN[22,25] with low $\sigma_{Tc}$, albeit over a narrow temperature range with $\Delta T_c/T_{c,AP} \ll 1$. The ultimate aim of absolute switching has not been achieved as yet to our knowledge.

Theoretically, absolute switching in a F/S/F structure requires a large proximity-induced magnetic exchange field ($h_{ex}$) in the S layer relative to its superconducting energy gap ($\Delta_0$) with a magnitude $h_{ex} > (\sqrt{2}/2)\Delta_0$ in the P-state[1]. It is well-established that $h_{ex}$ is proportional to the interfacial spin-mixing conductance $G_i$ (imaginary part) for constant $d_s$, and therefore, a large $h_{ex}$ corresponds to a large $G_i$[26–28]. Here, $G_i$ is a measure of the exchange field existing between the electrons in the non-magnetic metal and those in EuS and characterises the efficiency of F/N interfacial spin transport (see, e.g. refs. 28–30). For an F/S interface, this leads to a spin-splitting of the superconducting density of states (Fig. 1d)[31]. Pioneering experiments on Al/EuS structures (where Al is an S layer) were performed by Meservey, Tedrow and Moodera[31–36]. They demonstrated a splitting in the superconducting density of states[31] that corresponded to a magnetic field of more than 1 T. Non-superconducting experiments on EuS/Pt[28] and EuS/Graphene[37] structures also show evidence for large proximity-induced exchange fields, larger than 10 T in both Pt and Graphene. We note the recent experiments on Nb/EuS wires showing a supercurrent diode effect, which can be related to a large $h_{ex}$ in the Nb[38] and/or vortices[39].

Huertas–Hernando and Nazarov[40,41] theoretically proposed a modification of the F/S/F structures by inserting a normal metal layer (N) at the F/S interface as a means of achieving absolute switching. This N layer facilitates physical separation of the competing superconducting and magnetic order parameters and allows their careful control within N through superconducting and magnetic proximity effects. Here, we first report EuS/Nb/EuS structures with a superconducting switch efficiency $\Delta T_c/T_{c,AP}$ that can reach about 50%. In the next step, by inserting a 20-nm-thick heavy metal layer of Au at one interface (i.e. EuS/Au/Nb/EuS), we demonstrate a dramatic

enhancement of $\Delta T_c/T_{c,AP}$ reaching 1, achieving absolute switching[1]. The key to the enhancement of $\Delta T_c/T_{c,AP}$ is related to the interface chemistry and a larger proximity magnetic exchange field in Au due to a large $G_i$ at EuS/Au interface versus EuS/Nb interface. These results are obtained in extremely thin layers of 4-nm-thick Nb in which the superconducting state is preserved in the AP-state with the P-state showing no evidence of superconductivity down to 20 mK.

## Results

A set of NbO$_x$(3 nm)/EuS(30 nm)/Nb($d_{Nb}$)/SiO$_2$//Si, NbO$_x$(3 nm)/EuS(20 nm)/Nb(4 nm)/EuS(10 nm)/SiO$_2$//Si and NbO$_x$(3 nm)/EuS(20 nm)/Au(20 nm)/Nb(4 nm)/EuS(10 nm)/SiO$_2$//Si structures were prepared by electron-beam evaporation onto thermally oxidised silicon at room temperature (see 'Methods'). The 3-nm-thick top layer of NbO$_x$ is to protect the structure. The EuS layer has a predominantly in-plane magnetic anisotropy (see Supplementary Fig. 1), the 30-nm-thick EuS is insulating at room temperature with a contact resistance exceeding 10 GΩ (see Supplementary Fig. 2), and $d_{Nb}$ varies from 2 nm to 20 nm.

We first discuss the superconducting and magnetic properties of NbO$_x$(3 nm)/EuS(30 nm)/Nb($d_{Nb}$)/SiO$_2$//Si structures. Figure 1e shows the chemistry diagram of Nb (green), Eu (blue) and O (red) determined using a scanning transmission electron microscope, showing evidence for oxidation of the Nb capping layer. X-ray reflectivity measurements confirm the thickness of each layer (see Supplementary Fig. 3). Figure 1f shows $T_c$ vs $d_{Nb}$ for these structures, showing a decay in $T_c$ with relatively large values of $T_c$ of 0.2 K and 2.1 K for only 2- and 3-nm-thick Nb films, respectively. We define $T_c$ as the mid-point of the superconducting transition from a resistance vs temperature ($R(T)$) measurement. The current bias (1–10 μA) used to determine $T_c$ is sufficiently low and had no measurable effect on $T_c$ itself (see Supplementary Fig. 4). The width of the superconducting transition, $\sigma_{Tc}$, defined as the difference in temperature between 90 and 10% of the superconducting transition, is plotted in Fig. 1f showing relatively sharp transitions.

In Fig. 1g, h we have plotted the in-plane magnetic field trace of $R(H)$ of NbO$_x$(3 nm)/EuS(30 nm)/Nb(2 nm)/SiO$_2$//Si and NbO$_x$(3 nm)/EuS(30 nm)/Nb(3 nm)/SiO$_2$//Si unpatterned structures at temperatures across $T_c$. These show that near $T_c$ there is a local minimum in $R$ at the magnetic fields matching the coercive field ($H_c$) of the EuS layer, indicating recovery of superconductivity in the demagnetised state of EuS with the Ginzburg-Landau coherence length comparable to the magnetic domain size in the demagnetised state of EuS. The magnetisation vs in-plane magnetic field ($M(H)$) hysteresis loop in the top panel of Fig. 1h for the 30-nm-thick control sample of EuS shows that $H_c$ is about ±5.5 mT at 1.8 K. We note that the Curie temperature ($T_{Curie}$) of EuS is similar to the bulk value of about 16.6 K (see ref. 42 and Supplementary Fig. 5). The resistance minima in $R(H)$ match the $H_c$ of EuS of ±5.5 mT and are related to the recovery of superconductivity due to a reduction in $h_{ex}$ in Nb in the demagnetised state of EuS[21]. In the magnetised (single domain) state, $h_{ex}$ is maximal, thus maximising the suppression of $T_c$. The maximum measured shift in $T_c$ between magnetised and demagnetised states of EuS is about 150 mK for both the 2-nm- and 3-nm-thick Nb layers with the shift decreasing to zero as $d_{Nb}$ approaches the measured dirty-limit coherence length value of $\xi_s = 4.6$ nm (see Supplementary Figs. 6 and 7). These results demonstrate a robust magnetic proximity in superconducting Nb on a single layer of EuS.

We now discuss the performance of the superconducting switches. In Fig. 2a we have plotted the in-plane $M(H)$ loop for a NbO$_x$(3 nm)/EuS(20 nm)/Nb(4 nm)/EuS(10 nm)/SiO$_2$//Si structure at 4.2 K, which shows a differential switching around ±3 mT and ±6 mT, corresponding to different $H_c$ values of the two EuS layers. By sweeping the magnetic field from positive to negative directions, the relative magnetisation-alignment of the EuS layers changes from P to AP at about −3 mT. At −6 mT, the magnetisation of the harder EuS layer

switches, recovering a P-state. The extended data of the $T$-dependence of the $M(H)$ loops, remanence and $H_c$ of the two EuS layers are given in Supplementary Figs. 8 and 9. The bottom panel of Fig. 2a shows the corresponding $R(H)$ in the superconducting transition at 4.2 K: in the P-state, there is a finite resistance in the normal state with superconductivity recovered in the AP-state, which translates to an infinite magnetoresistance, confirming a full superconducting switch effect. We define magnetoresistance as $(R_{H=0} − R_{H=Hc})/R_{H=Hc}$. We note that the switching fields in $R(H)$ do not perfectly match the switching fields in $M(H)$, possibly due to a canted surface magnetic moment on EuS, similar to $R(H)$ scans reported in EuS/Al/EuS structures[21]. We also note that a small asymmetry of about ±1 mT occurs in $R(H)$ due to the trapped flux from the superconducting coils of the cryostat.

In Fig. 2b we have plotted the zero-field $T$-dependence of $R_{AP}$ and the $T$-dependence of the normalised resistance mismatch between P- and AP-states derived from individual $R(H)$ scans at each temperature, i.e. $(R_P(T) − R_{AP}(T))/R_N(T) = \Delta R(T)/R_N(T)$, where $R_N(T)$ is the resistance in the normal state. Selected $R(H)$ scans at temperatures across $T_c$ are shown in Supplementary Fig. 12, and zero-field $R_{AP}(T)$ is obtained using the method described in Supplementary Fig. 13. From these measurements we obtain a superconducting switch efficiency of $\Delta T_c/T_{c,AP} = 0.3$ in NbO$_x$(3 nm)/EuS(20 nm)/Nb(4 nm)/EuS(10 nm)/SiO$_2$//Si. An efficiency of $\Delta T_c/T_{c,AP} = 0.5$ is determined for the same structure in Supplementary Figs. 10, 13 (Noted as Device 3). We note that the Nb interlayer is optimised to 4-nm-thick, where the localisation effect becomes noticeable when the Nb interlayer thickness is further reduced to 3 nm, as described in Supplementary Fig. 14.

To investigate boosting $\Delta T_c/T_{c,AP}$ of the F/S/F structure by inserting a single N interlayer[40,41], we fabricated a NbO$_x$(3 nm)/EuS(20 nm)/Au(20 nm)/Nb(4 nm)/EuS(10 nm)/SiO$_2$//Si structure with the heavy metal layer of Au at one Nb/EuS interface. The top panel of Fig. 2c shows the in-plane $M(H)$ loop of the structure at 1.8 K which closely matches the equivalent structure without Au in Fig. 2a. From the normalised $R(T)$ (green curve, Fig. 2d) we estimate that $T_{c,AP}$ is about 1.86 K. The additional suppression of $T_c$ most likely arises from the proximity of the thin Nb layer with the 20-nm-thick Au layer. Remarkably, in this hybrid structure, we observe an infinite magnetoresistance and a normal state resistance in the P-state down to the lowest measurable temperature of 20 mK. The ability to maintain a non-superconducting normal state for P magnetisations down to 20 mK demonstrates absolute switching.

For comparison, in Fig. 3 we have plotted the superconducting switch efficiency $\Delta T_c/T_{c,AP}$ values in this study to equivalent structures in the literature involving transition metal ferromagnets or rare-earth ferromagnets. EuS/Nb-based structures show $\Delta T_c/T_{c,AP}$ efficiencies that exceed values measured in equivalent structures, including EuS/Al[11–23]. We note that in ref. 21 $T_{c,P}$ was undetectable down to 1 K, but lower temperature data are not provided, and so an absolute spin-valve effect cannot be concluded.

## Discussion

The enhancement of $\Delta T_c/T_{c,AP}$ due to the heavy metal layer of Au is, at first glance, unexpected. Firstly, Au has relatively strong spin-orbit coupling, which should smear the induced spin splitting of the superconducting density of states in Nb due to the EuS, thereby countering the suppression of $T_c$ caused by the proximity-induced magnetic exchange field interaction[43]. Therefore, one would in fact expect a smaller contrast between $T_{c,P}$ and $T_{c,AP}$ in the EuS/Au/Nb/EuS structure. Secondly, theory predicts that the proximity exchange field induced in S ($h_{ex}$) is inversely proportional to the layer thickness (i.e. $h_{ex} = \kappa_{int}/d$)[44–47]. $\kappa_{int}$ is a parameter quantifying the interfacial exchange field related to $G_i$ via $G_i \approx \pi G_0 N_F \kappa_{int}$, where $G_0$ is the conductance quantum and $N_F$ is the Fermi level density of states per spin[48]. If we assume that $\kappa_{int}$ at the EuS/Nb interface equals to the EuS/Au interface as indicated in Supplementary Fig. 16, the addition of Au should

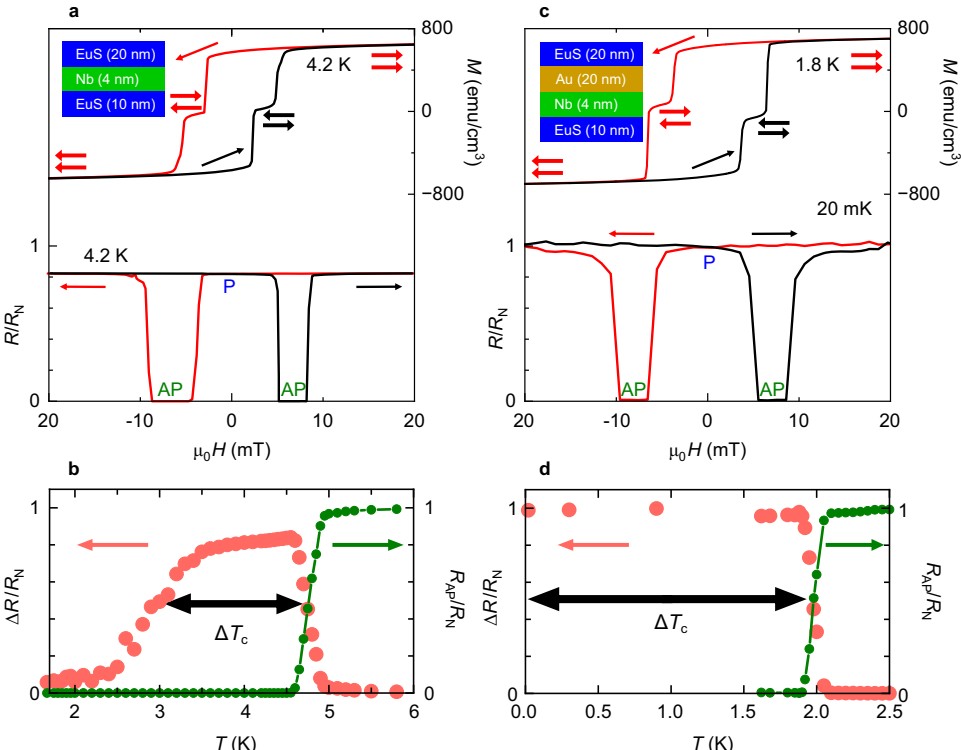

**Fig. 2 | Superconducting switch performance with or without a heavy metal interface interlayer. a** $M(H)$ (right axis) and $R(H)$ (left axis) from an unpatterned $NbO_x(3\,nm)/EuS(20\,nm)/Nb(4\,nm)/EuS(10\,nm)/SiO_2//Si$ structure (Device 1) at 4.2 K. Single arrows indicate the magnetic field sweep directions and double arrows represent possible magnetisation directions of the top and bottom EuS layers. Top left inset: schematic cross-section of the structure. **b** $R_{AP}(T)/R_N(T)$ (in green) and $\Delta R(T)/R_N(T)$ (in pink) of each $R(H)$ scan. **c** $M(H)$ at 1.8 K (right axis) and $R(H)$ at 20 mK (left axis) of an unpatterned $NbO_x(3\,nm)/EuS(20\,nm)/Au(20\,nm)/Nb(4\,nm)/EuS(10\,nm)/SiO_2//Si$ (Device 2). Top left inset: schematic cross-section of the structure. **d** $R_{AP}(T)/R_N(T)$ (in green) and $\Delta R(T)/R_N(T)$ (in pink) of each $R(H)$ scan, showing absolute switching with $\Delta T_c/T_{c,AP}$ equal to 1 (approximately). Data below 1 K are for the same structure measured in a different cooling in a dilution fridge.

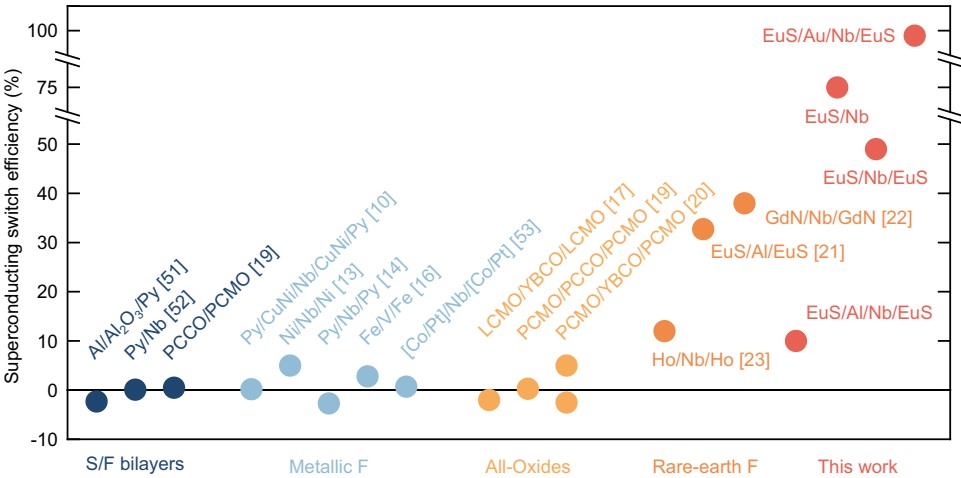

**Fig. 3 | Literature survey of superconducting switch efficiencies for F/S/F structures with different materials combinations, including transition metal ferromagnets and *f*-orbital ferromagnets.** PCMO is $Pr_{0.8}Ca_{0.2}MnO_3$, PCCO is $Pr_{1.85}Ce_{0.15}CuO_4$, LCMO is $La_{0.7}Ca_{0.3}MnO_3$ and YBCO is $YBa_2Cu_3O_7$.

suppress the effective exchange interaction by increasing the distance between the EuS layers thereby reducing the value of $\Delta T_c/T_{c,AP}$.

Instead, by inserting a layer of Au we see a strong enhancement of $\Delta T_c/T_{c,AP}$. This enhancement likely results from an increase in the exchange coupling at the EuS/Au interface relative to the EuS/Nb interface. This is, in principle, not surprising, since the value of $\kappa_{int}$ is sensitive to microscopic details of the interface, including atomic structure and lattice mismatch[48,49]. Indeed, a large interfacial exchange

coupling at the EuS/Au interface has been reported elsewhere[50]. Moreover, the addition of the heavy metal layer Au may partially suppress $T_c$ via the inverse proximity effect as indicated in Supplementary Fig. 17, favouring the suppression of superconductivity and hence reducing the critical field. This may add to the suppression of superconductivity in the P state.

For a more quantitative understanding, we have calculated the $T_c$ of the different F/N/S/F structures (where F is an insulator) using the

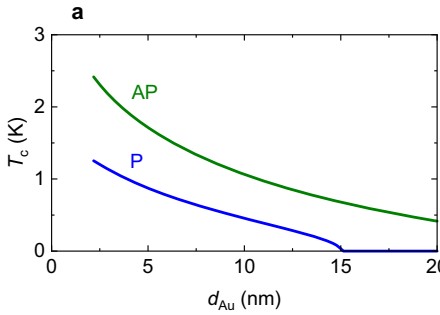

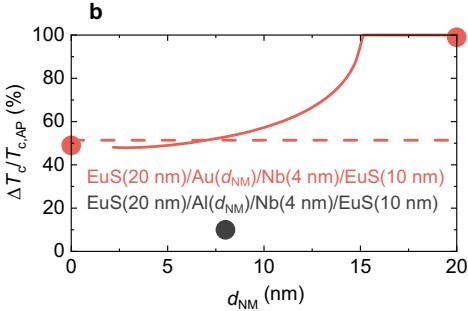

**Fig. 4 | Calculated superconducting switch efficiency of EuS/Au($d_{NM}$)/Nb(4)/ EuS structures. a** $T_{c,P}$ (in blue) and $T_{c,AP}$ (in green) as a function of $d_{Au}$. **b** $\Delta T_c/T_{c,AP}$ as a function of $d_{NM}$. For optimised proximity-induced magnetic exchange fields of $\kappa_{EuS/Au} = 1.5$ meV·nm at the EuS/Au interface and $\kappa_{EuS/Nb} = 1.2$ meV·nm at the EuS/Nb interface, absolute switching is expected for $d_{Au} \geq 15$ nm (Solid line). The dashed line in (**b**) corresponds to $d_{Au} = 0$. Dark grey data with $d_{Al} = 8$ nm indicates the control structure involved inserting an 8-nm-thick Al spacer in a EuS(20 nm)/Al(8 nm)/Nb(4 nm)/EuS(10 nm) structure. The superconducting switch efficiency decreases to 10%.

Usadel framework based on the quasiclassical Green's functions. Here, we present the main results related to the experiment, with details of the model available in the Supplementary Note 2.

In Fig. 4a we have plotted the calculated $T_c$ of the EuS/Au/Nb/EuS structure vs Au layer thickness ($d_{Au}$) in the P- (in blue) and AP- (in green) magnetic states. For $d_{Au} \geq 15$ nm, we are able to obtain a complete suppression of $T_{c,P}$ with $T_{c,AP}$ nonzero for an optimised induced exchange coupling with $\kappa_{EuS/Au} = 1.5$ meV·nm and $\kappa_{EuS/Nb} = 1.2$ meV·nm, equivalent to $G_i = 2.15 \times 10^{13}\ \Omega^{-1}\ m^{-2}$ at EuS/Au and $G_i = 1.6 \times 10^{13}\ \Omega^{-1}\ m^{-2}$ at EuS/Nb interfaces[50], where $G_i$ for EuS/Au is larger than for EuS/Nb. Our estimates of $G_i$ are similar to values reported for EuS/Pt ($G_i = 7 \times 10^{12}\ \Omega^{-1}\ m^{-2}$)[28] in which the EuS and Pt layers are deposited in a separate vacuum system, compromising the interface quality, which reduces $G_i$. Furthermore, our $G_i$ for EuS/Au is also similar to YIG/Au ($G_i = 1.73 \times 10^{13}\ \Omega^{-1}\ m^{-2}$)[29].

Figure 4b shows the dependence of the maximum superconducting switch efficiency vs $d_{Au}$. The dashed line is for $d_{Au} = 0$. This value differs from the $d_{Au}$-0 nm limit (highlighted by the solid line) due to the finite interface resistance at the Nb/Au interface and the different exchange coupling strengths at the EuS/Au interface. If the Au is thick enough ($d_{Au} \geq 15$ nm), $T_{c,P}$ is suppressed for all temperatures, achieving an absolute superconducting switch.

To further investigate the role of the interfacial layer in enabling the absolute superconducting spin-valve effect, we performed a control experiment in which the heavy metal Au was replaced with the lighter metal Al in a NbO$_x$(3 nm)/EuS(20 nm)/Al(8 nm)/Nb(4 nm)/EuS(10 nm)/SiO$_2$//Si structure. Al has negligible spin–orbit interaction[54], with $\kappa_{EuS/Al}$ estimated to be 0.7–0.9 meV·nm[21,28,31] and being lighter than Au lacks the strong interfacial effects typically associated with heavy elements. These structures exhibited a significantly reduced switching efficiency, with $\Delta T_c/T_{c,AP}$ dropping to ~10% (see Figs. 3 and 4b and Supplementary Fig. 15). This result highlights the important roles of both the interfacial chemistry and interfacial exchange parameter $\kappa$ in facilitating the efficient transmission of the magnetic proximity effect from EuS into the superconducting Nb layer.

The mechanism underlying this proximity exchange transmission effect is related to $s$–$d$ orbital exchange interaction at the F/metal interface, where localised $f$-electron moments in EuS couple to the itinerant electrons in the adjacent normal metal layer. This coupling transmits the exchange field across the interface and decays with increasing normal metal thickness[55]. The strength of this interaction is dependent on interface quality and the specific materials involved, requiring an intimate electronic overlap between the EuS $f$-electrons and the 's states' of the adjacent metal. While the magnitude of the exchange field itself does not depend directly on spin–orbit coupling–

and has been reported to be similar for both Al[31] and heavier metals such as Pt[28]—our results suggest that the enhanced $G_i$ and interface-induced spin-mixing conductance in Au are key to achieving strong magnetic proximity effects and, ultimately, the absolute switching observed in EuS/Au/Nb/EuS structures.

In summary, we have demonstrated absolute switching in a EuS/Au/Nb/EuS structure. The switch effect is boosted by the large proximity exchange field induced Au vs Nb, which enables absolute on/off switching of superconductivity. The results could create interest in exploiting these effects. For example, a large $\Delta T_c/T_{c,AP}$ ratio is key towards the development of non-volatile superconducting random access memory. Wires which can be controllably switched between superconducting and non-superconducting states are already used in a variety of applications which range in scale from those in persistent mode superconducting magnets, to small scale devices to break SQUID pick-up loops in NMR systems so that large currents are not induced during field ramps, but all current devices are thermally controlled, so that a heater drives the device above $T_c$. A magnetic switch would eliminate the continuous heat load required to hold a thermal switch open (which can be a significant load on the cryogenic system), albeit requiring careful design to eliminate stray field effects for certain applications.

## Methods
### Film growth
Thin-films are deposited onto 5 mm × 5 mm area precleaned thermally oxidised silicon substrates at room temperature in a custom-built ultra-high vacuum electron-beam evaporator with a base pressure of $5 \times 10^{-9}$ mbar. EuS is evaporated directly from EuS powders with an average diameter of less than 44 μm. All materials are evaporated with a growth rate of ~1 nm·min$^{-1}$. We investigate NbO$_x$(3 nm)/EuS(30 nm)/Nb($d_{Nb}$), NbO$_x$(3 nm)/EuS(20 nm)/Nb(4 nm)/EuS(10 nm) (Device 1 and 3), NbO$_x$(3 nm)/EuS(20 nm)/Au(20 nm)/Nb(4 nm)/EuS(10 nm) (Device 2), NbO$_x$(3 nm)/EuS(20 nm)/Nb(3 nm)/ EuS(10 nm) (Device 4) and NbO$_x$(3 nm)/EuS(20 nm)/Al(8 nm)/Nb(4 nm)/EuS(10 nm) (Device 5) structures. The 3-nm-thick top layer of NbO$_x$ is to protect the structure. The different EuS layer thicknesses ensure different coercive fields for independent magnetisation switching between P and AP state. The central Nb layer thickness is optimised to be 4 nm, which is thinner than the dirty-limit superconducting coherence length of bulk Nb ($\xi_s$) but thick enough to ensure a relatively sharp superconducting transition width.

### Magnetic measurements
The magnetic moment vs magnetic field measurements are performed in a Quantum Design Magnetic Property Measurement System equipped

with a vibrating sample magnetometer superconducting quantum interference device (SQUID). The system can apply up to 7 T using a superconducting magnet with a magnetic moment sensitivity of about $10^{-8}$ emu.

## Electron microscopy characterisation

Cross-sectional lamellae are prepared using a Dual Beam focused ion beam microscope. Low/high-resolution annular dark field imaging and X-ray energy-dispersive spectrum imaging are carried out using an aberration-corrected (probe) Thermo Fisher Themis-Z operated at an accelerating voltage of 200 kV. The electron energy-loss spectroscopy (EELS) data are acquired on a Thermo Fisher Titan3 G2 60-300 S/TEM at 300 kV, equipped with a high-brightness field emission electron gun, a monochromator and a dual-EELS spectrometer. The pixel dwell time is 0.5 s with ×16 sub-pixel scanning. The EELS data are analysed using Gatan Digital Micrograph software to obtain elemental quantification from deconvolved and background-removed Nb-M, Eu-N, S-L and Si-L edges.

## X-ray reflectivity measurements

Thickness calibration is performed using X-ray reflectometry with a Brucker D8 diffractometer using copper K-α radiation with a wavelength of 1.54 Å. From Kiessig fringes, we estimate layer thicknesses using the Leptos software and a genetic algorithm of approximation. The simulation model corresponded to the structure of the original sample and the measurement conditions used in the experiment in each case.

## Superconducting electrical measurements

Low temperature current-voltage ($I(V)$) measurements are performed using a four-terminal electrical setup. Measurements above 1 K are performed in a cryogen-free system (Cryogenic Ltd) with an in-plane magnetic field and temperature stability of at least 10 mK. Measurements in the mK range are performed in Oxford Instruments Triton 200 Dilution Refrigerator with 6-1-1 vector magnet and 20 mK electron temperature. The $I(V)$ characteristics are measured using a current-bias of 1–10 μA and AlSi ultrasonically-bonded contacts on the thin-film multilayers via 4-probe in line.

## Data availability

Data sets generated during the current study are available from the corresponding author on request.

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

## Acknowledgements

Low-temperature transport and volumetric magnetisation measurements were supported by Cambridge Royce facilities grant EP/P024947/1 and Sir Henry Royce Institute recurrent grant EP/R00661X/1. J.W.A.R. and H.M. acknowledge support from the Henry Royce Institute for Advanced Materials through the Equipment Access Scheme. H.M. and J.W.A.R. acknowledge funding from the EPSRC through International Network and Programme Grants (No. EP/P026311/1 and No. EP/N017242/1). G.Y. acknowledges funding from the National Key Research and Development Programme of China (No. 2022YFA1402604) and the Natural Science Foundation of China (52201200). The electron microscopy work was primarily supported by DARPA under Grant No. D18AP00008. D.W.M. acknowledges support from the Centre for Emergent Materials at the Ohio State University, a National Science Foundation Materials Research Science and Engineering Centre (Grant No. DMR-2011876). B.W. acknowledges support from the Presidential Fellowship of the Ohio State University. Electron microscopy experiments were performed at the Centre for Electron Microscopy and Analysis at the Ohio State University. A.H. acknowledges funding from the University of the Basque Country (Project PIF20/05) and Research Council of Finland through the Finnish Quantum Flagship (project No. 359240). F.S.B. and A.H. acknowledge financial support from Spanish MCIN/AEI/10.13039/501100011033 through projects PID2020-114252GB-I00, PID2023-148225NB-C31 and TED2021-130292B-C42, the Basque Government through grant IT-1591-22 and the European Union's Horizon Europe research and innovation programme under grant agreement No 101130224 (JOSEPHINE). S.I. is supported by the Research Council of Finland (project No. 355056). S.K. acknowledges funding from the JST FOREST Grant (No. JPMJFR212V).

## Author contributions

J.W.A.R. had the original idea of the project and developed it with G.Y. and H.M. The samples were prepared by H.M. and G.Y. with the help of G.P.M., K.O., S.K. and N.S. Electrical transport measurements were carried out by H.M., G.P.M., Y.L. and G.Y. X-ray diffraction was performed by I.A. The electron microscopy characterisation was performed by B.W. and D.W.M. The model calculation was performed by A.H., S.I. and F.S.B. N.B. and L.F.C. revised the manuscript. All authors discussed the results and commented on the manuscript, which was primarily written by H.M., G.Y. and J.W.A.R.

## Competing interests
