## [Peer Review file · Nature Communications]

Realisation of de Gennes' Absolute Superconducting Switch with a Heavy Metal Interface

Corresponding Author: Professor Jason Robinson

Version 0:

Reviewer comments:

Reviewer #1

(Remarks to the Author)

I have read the manuscript "Realisation of de Gennes' Absolute Superconducting Switch with a Heavy Metal Interface" by Hisakazu Matsuki and collaborators, submitted to Nature Communications for consideration.

The paper reports an absolute superconducting switch effect in EuS/Au/Nb/EuS structures, where the T_c is strongly modulated by the alignment (parallel, P vs antiparallel, AP) of the magnetic moments of the EuS ferromagnetic insulating layers. The enhanced T_c modulation found in structures with an Au interlayer is discussed in terms of enhanced spin mixing conductance, namely of the exchange field, at the EuS/Au interface as compared to the EuS/Nb case.

The paper contains an interesting "new" result on an "old" physical scenario enabled by the successful growth of ultrathin EuS layers in metallic multilayers. The experimental evidence shown for the absolute switch effect is compelling, and the result may be of interest for advanced quantum sensing. Before recommending acceptance, I need authors to distil the explanation.

The critical temperature of the stack is modelled with the quasi classical Green's functions technique. Authors argue that the absolute switch effect results from an enhanced exchange field at the Au/EuS interface. The exchange field is expressed in terms of the interfacial interaction parameter $\kappa_{\text{EuS/N}}$ and the thickness of the layer $d_{\text{S/N}}$ as $h_{\text{S/N}}(d) = \kappa_{\text{int,S/N}}/d_{\text{S/N}}$.

Although a slightly larger $\kappa_{\text{EuS/N}}$ is found in the case of Au interface ($\kappa_{\text{EuS/Nb}} \sim 1.2 \text{ meV nm}$ and $\kappa_{\text{EuS/Au}} \sim 1.5 \text{ meV nm}$), the rather thick (20 nm) Au layer used in the case of absolute switch, should decrease the exchange field to values well below the case of pristine EuS/Nb interface. At the same time, it is found that the T_c of the Au/Nb (4nm) stack is substantially depressed by S-N proximity effect from $T_c=5.5\text{K}$ (no Au) to $T_c = 1.86 \text{ K}$ (20 nm Au). Can authors discuss the relative importance of modified exchange field and proximity suppressed T_c in the samples with Au interlayer? Can both effect be separated (may be with an ultrathin insulating layer in between of EuS and Au)?

Another point which needs further discussion is the effect of the magnetic domain state of the EuS on the absolute switch effect. A single EuS layer in the domain state produces also substantial T_c modulation and magnetoresistance effect. This is addressed in Figure 1 g and h for the case of samples with no Au interlayer. In this regard, it is proposed that the effect is maximized when superconductor layer thickness is below the coherence length, however, the size of magnetic domains is not quoted, neither its effect is discussed. Can authors describe the domain state of EuS? Is it modified in presence of the superconductor? How about magnetic anisotropy? What is the evidence of the canting of magnetic moments mentioned in the text? It would be helpful having data on magnetometry and transport of samples with a single EuS layer AND Au interlayer. On general grounds one expects that the exchange field will be cancelled if domain size is below the superconductor coherence length. It is probably too small (4nm) to be the case. What is then the reason for the magnetoresistance suppression at coercivity in samples with a single EuS layer?

Minor: Please revise sentence on lines 217- 219.

Reviewer #2

(Remarks to the Author)

The article discusses the detection of the absolute superconducting switch effect in hybrid magnetic/superconducting structures by the use of a heavy metal Au layer.

The results are very interesting both from a purely scientific point of view and for their potential technological interest.

The manuscript is well written and presented, the graphics are clear, and the supplementary material is sufficient to back the claims.

I would recommend this article for publication with minor revisions, if the authors answer the minor concerns stated below:

1. Figure 1.e. The diagram shows an NbOx/EuS/Nb structure, while the caption mentions Nb/EuS/Nb. As can be read in the text, the top Nb layer is for protecting the structure, but the authors should be consistent and the structure should be the same in the Figure, caption, and main text.

2. Also, the caption refers to the image as a "Chemistry diagram". The authors should add that STEM was used to produce the image.

3. The authors should include some justification as to why an Nb protection layer was used, instead of e.g. AlOx or some other capping material.

4. Figure 2a, the AP states do not seem entirely symmetrical, negative fields produce a wider AP state. Is this due to the canted surface magnetic moment on EuS?

5. Fs hasn't been defined.

174 For comparison, in Fig. 3 we have plotted the superconducting switch efficiency $\Delta T_c/T_c, AP$ values in this study
175 to equivalent structures in the literature involving transition metal Fs or rare-earth Fs.

6. Tc of the EuS/Au/Nb sample is reduced down to 2K, compared to the sample with no Au.

The authors state:

"Moreover, the addition of the heavy metal layer Au may partially suppress Tc via the inverse proximity effect, favouring the suppression of superconductivity, and hence reduce the critical field. This may add to the suppression of superconductivity in the P state."

It seems to me this would be easily tested by the authors with a control sample Au/Nb with no EuS.

Reviewer #3

(Remarks to the Author)

This manuscript presents a study of superconducting spin valves with Nb as the superconductor, EuS as the ferromagnetic insulator and the novel introduction of Au layer between one of the Nb/EuS interfaces. Authors argue that Au increases the spin mixing conductance and possibly enhances the proximity induced exchange fields in Nb – which results in the Nb perpetually remaining in the metallic state in the parallel orientation of both the EuS layers. Authors term the as the 'Absolute spin valve' as the spin valves turn superconducting only for anti-parallel orientation of the EuS layers. Authors also claim in Figure 3, that their present work represents the highest magnitude of $\Delta T_c/T_c$'s (defined as Superconducting switch efficiency or SCE) ever observed and probably is close to 100%.

While it is possible that Au may indeed enhance magnetic proximity coupling in Nb; I have few glaring concerns regarding these claims which make this paper unworthy of publication. These are stated below:

1) Claims made in Figure 3 and the numbers presented are simply not correct. Please refer to discussions of Figure 4a of reference 18. It is clear that the SCE in that device is at least 35% ($\{1.55 - 1\}/1.55$). Similar is the case for reference 20, where Figure 3d show that SCE is at least 85% ($\{2 - 0.3\}/2$). These numbers are massively understated as approximately 20% and 40% in the manuscript.

2) In both references 18 and 20, the authors achieved a similar scenario as in this manuscript – that the parallel state Tc was below the measurable lowest temperature in their respective measurement systems. While in Ref. 18, it is stated to be 1K, in Ref. 20 it is probably 300mK. It is very much possible that in both cases, absolute spin switches were already achieved. This is especially the case in Ref. 18, where authors explicitly estimate the exchange field to be approximately 13meV, which translates to several tens of Teslas.

3) More importantly, I have a major concern regarding the fact that the SCE increases with increasing Au layer thickness (Figure 4b). This data can have an alternative interpretation, when Figure 4a is taken into consideration simultaneously.

This can be the Metal/Superconductor proximity effect in action that is weakening superconductivity in Nb with increasing Au thickness. This is because with increasing Au thickness, Tc (AP) is seen to be monotonically decreasing, as it should in a Nb/Au bilayer. The Tc (AP) is always higher than Tc (P), because the induced exchange fields due to the top and bottom EuS layers cancel each other in the AP orientation. If one assumes that there is some asymmetry of induced exchange fields at the top and bottom interfaces; the cancellation should be more due to EuS/Al/Nb interfaces (as per the authors argument that exchange fields in such cases increase), and hence Tc (AP) of the high thickness Au layers should not decrease by more than 1K as shown in Figure 4a. Hence, in the case that this observation is just a Nb/Au proximity effect, the massive Tc

(AP) lowering is readily explained, and low/unobservable T_c (P) is due to adding up of exchange fields in the P orientation on an already metal proximity weakened superconductor.

4) Considering the above scenario, one must reflect on the question that what could be an alternative simpler realisation of the absolute spin valve effect. One can imagine, that if instead of inserting the Au layer, one can just reduce the Nb thickness in EuS/Nb/EuS multilayer structure. The reduced Nb thickness would result in reduced T_c (AP) and lower thickness of Nb would result in enormously high average exchange fields in the Nb layer when the EuS layers are parallelly aligned. For some Nb thickness not too much less than 4nm, the P orientation induced exchange fields will far exceed the paramagnetic limit of thin film Nb and result in non-observation of superconductivity at all temperatures.

Considering the fact, that the authors seem to have exemplary high-quality sample growth facilities, it would be a reasonable experiment to try out. In other words, the requirement of Au layer which is proposed as the key element for realisation of an absolute spin switch, may altogether be bypassed.

Taking all these factors into account, the manuscript in its current form is not fit for publication. The introduction of Au layer and its implications especially with regards to consideration of Metal/Superconductor proximity effect and associated lowering of T_c needs to be discussed thoroughly. Moreover, there can be much simpler controlled schemes for realising the same phenomena. Hence, at this moment this looks like an incremental piece of work, which needs further experimental evidence and analysis to bolster the claims made.

Version 1:

Reviewer comments:

Reviewer #1

(Remarks to the Author)

I have read the revised manuscript "Realisation of de Gennes' Absolute Superconducting Switch with a Heavy Metal Interface" by Hisakazu Matsuki and collaborators, and authors' rebuttal to the criticism in the first round of reviews.

Authors have satisfactorily addressed my criticism and modified the manuscript accordingly, making, in my view, a more solid case of their absolute spin switch effect.

To my question of the relative importance of exchange field and proximity effect in suppressing T_c , authors have clarified the importance of the exchange interaction parameter. A new figure S14 makes clear that for increased strength of the interfacial exchange interaction thicker Au interlayers would be necessary to observe the absolute spin switch effect. This clarification helps grasping the effects underpinning the absolute spin switch effect.

To my question on the role of domain size in depressing the critical temperature, authors have made clear that the important length scale is the (diverging at T_c) Ginzburg Landau coherence length. Authors have also considered my questions on domain size and anisotropy and included new magnetometry data.

I maintain my view on the interest of the large T_c modulation in EuS/Au/Nb/EuS structures by magnetic alignment of the EuS ferromagnetic layers. The result could attract readers working on superconductivity but also readers from electronics and engineering as the reported effect could be of interest for quantum sensing.

My recommendation is to publish this manuscript in its current revised form.

Reviewer #2

(Remarks to the Author)

The authors' answers and changes to the manuscript are satisfactory and I would be happy to recommend the article for publication.

Reviewer #3

(Remarks to the Author)

I have gone through the new version of the manuscript, and have carefully read through all the rebuttal arguments provided by the authors. Following which, my assessment is the following:

1) Is absolute spin switching conclusively proven?

Yes

2) Is absolute spin switching due to a heavy metal interface theoretically possible as per theory presented in supplementary section?

Yes

3) Most importantly: Is absolute spin switching due to a heavy metal interface experimentally proven unambiguously in this manuscript?

No. The authors do not prove conclusively that it's the modified exchange field and not the Nb/Au proximity effect that causes the above.

The reasons are stated below:

a) The entire experimental evidence argument almost stands on this one point which is quoted from the rebuttal: "Our analysis suggests that while both effects are essential to achieving the absolute switch effect, the modified exchange field plays a more decisive role. This conclusion is underpinned by a direct comparison of the change in ΔT_c between Fig. 2b ($\Delta T_c \sim 1.6$ K without Au) and Fig. 2d ($\Delta T_c \sim 2$ K with Au) in the main text."

This statement/observation is wrong because of reasons stated below:

- In Fig 2d. The T_c AP is less than 2K (approximately around 1.75). Hence ΔT_c cannot be 2K and is much lesser. Reasons are stated in points below.

- The plot in 2d is wrong, as the green line and red dots must meet each other at the 0.5 mark on either y axes (as is the case in Figures 2b, S10 and S13)

Suggestion to authors - this mistake could be due to improper selection of data of temperature sensors in the dilution fridge. Typically for the low temperature ranges, dilution refrigerators will have a sensor (mostly RuO₂) that is more accurate below approximately 1.4K; and another (mostly cernox) which is more accurate above 1.5K. It may be the case, that for the 20nm Au sample, all RvsH data (above and below 1.4K) was recorded with RuO₂ sensor. And RvsT data was recorded with cernox. And hence this mismatch. Between 1.6 to 2K and above, the two sensors are known to have a substantial mismatch.

- The above mismatch also shows up in RvsT shown in Fig. 2d and RvsH in second from bottom figure of middle panel in Fig. S11. As per the RvsT, there should be no zero-resistance state in RvsH at 1.8K. So, this is due to some error in data collection/temperature sensor selection.

- Fig. S15 is very important, as it shows the effect of proximity of 20nm Au on 4nm Nb. From this plot the T_c of such a bilayer seems to be somewhere in the vicinity of 1.5K. Therefore, how is it possible that when this same bilayer is sandwiched by two magnetic EuS layers – the T_c AP (as per DeGennes, in the AP state, the exchange field should ideally be zero) can at most be 1.5K. In reality, the T_c AP of the sandwiched structure is almost always observed to be less than that without magnetic sandwich T_c . So, it is possible that when the EuS/20nm Au/4nm Nb/EuS sample was grown, the thickness of Nb is definitely greater than the Nb grown for 20nm Au/ Nb bilayer shown in Fig. S15. Therefore, it seems that the comparison in Fig. 2a and Fig. 2d is not a like to like Nb thickness comparison.

- Fig. 2b is also suggestive of broadening of T_c P. Hence, if any alternative definition of ΔT_c is applied (other than comparison of 50% of transition point); the ΔT_c will be much larger and can even go to a maximum of 2K. When the comparison is so tight, this point needs to be thought of very carefully before attributing the observed effect more to heavy metal interface than to Au/Nb proximity effect.

Hence, due to all of the above inconsistencies/concerns; as of now, this rebuttal of the authors does not stand.

b) A concerning but relatively minor point in comparison to the above.

"In our system, when the Nb thickness is reduced below 4 nm, localisation effects become noticeable, indicated by an initial resistivity increase before the superconducting transition shown in the top panel of Fig. S13. In such cases, the coherence length of highly disordered Nb could be further reduced below its thickness. As shown in the middle and bottom panels of Fig. S13, we do observe an enhanced superconducting switch in the NbOx(3 nm)/EuS(20 nm)/Nb(3 nm)/EuS(10 nm)/SiO₂/Si device prepared under identical growth conditions, but with lower ΔT_c (~ 0.5 K) and $\Delta T_c/T_c, AP$ ($\sim 12\%$) compared to the NbOx/EuS(20 nm)/Au(20 nm)/Nb(4 nm)/EuS(10 nm)/SiO₂/Si device."

- I do not agree with these arguments due to apparently contradictory data shown by the authors themselves in Fig. 1 of the manuscript. Fig. 1f, 1g and 1h clearly shows that bilayers NbOx(3 nm)/EuS(30 nm)/Nb(2 nm)/SiO₂/Si and / NbOx(3 nm)/EuS(30 nm)/Nb(3 nm)/SiO₂/Si show superconductivity at 1K and 1.8K, and proximity effect in terms of domain wall superconductivity is clearly visible. Hence, I find it hard to believe how and why a NbOx(3 nm)/EuS(20 nm)/Nb(3 nm)/EuS(10 nm)/SiO₂/Si cannot be made properly and measured.

- It sounds highly improbable that the effective bilayer of EuS(30 nm)/Nb(3 nm) shown in Fig 1h should have a lower T_c than the effective trilayer EuS(20 nm)/Nb(3 nm)/EuS(10 nm) shown in Fig. S13. It is likely that the sample whose data is shown in Fig. S13, is not a 3nm Nb sample, but one with Nb thickness is much higher. Otherwise, all of the data taken together – is simply inconsistent.

- As a side note. What is really the point of Fig. 1 in this paper?

This figure has nothing to do with a DeGennes spin valve. The fact that exchange field exists in EuS/Nb samples, is also evident from trilayers. The RH plots shown in Fig. 1 involves completely different physics of domain wall superconductivity which is discussed briefly and is kind of irrelevant in the context of this paper.

Other points:

- “For reference 18 (now reference 21), the SCE of EuS/Al/EuS has been adjusted to $(1.55-1)/1.55 = 35\%$. Regarding reference 20 (now reference 22), the reviewer calculated an SCE of 85% using $T_{c,P} = 0.3$ K and $T_{c,AP} = 2$ K from Fig. 3d. However, the $T_{c,P} = 0.3$ K value was measured with an external field of 80 mT, which further suppresses $T_{c,P}$. To avoid this mistake, we note that our measurements of $R(T)$ are performed in zero magnetic field. Hence, the correct SCE from reference 20 should be about 40 %, determined from the right inset of Fig. 2, where zero field $R(T)$ data are presented.”

Agreed.

- “It is unfortunate the authors did not investigate $T_{c,P}$ down to mK temperatures but it should not be a ground to raise doubts on the novelty of our new observations going down to 20 mK.”

Agreed. The value of exchange field of 13.6meV in now Ref. 21 actually is suggestive that since it is so much higher than gap value of Al – even if it is measured down to dilution temperatures, it would probably be still non-superconducting.

Nevertheless, the reasons for not agreeing with the authors analysis is not based on what has/has not been done before; but based purely on the data presented in this manuscript and its supplementary information.

So in light of the above observations (mainly the ones stated in point a)), I am certain of data inconsistencies in this manuscript; which might have led to fallacious comparisons and therefore inaccurate conclusions. This manuscript is therefore currently not fit for publication.

Version 2:

Reviewer comments:

Reviewer #3

(Remarks to the Author)

Having carefully gone through the rebuttal letter by the authors, and after considering the errors corrected from the last version of manuscript; I am still not fully convinced by the experimental results that the authors show to prove their claim – that the heavy metal interface provided by EuS/Au contact is the most important determining factor for observation of absolute spin switch effect. In effect, the only experimental evidence that is shown is that the ΔT_c of a 4nm Nb spin valve without and with Au remains roughly the same (approximately little less than 2K).

The rest is all about apparently compelling arguments. I give two examples:

The authors use their theory (over which I have no doubts) to estimate $\kappa_{EuS/Nb} \sim 1.2$ meV \times nm and $\kappa_{EuS/Au} \sim 1.5$ meV \times nm (presumably from fits to experimental data). These fitted parameter values are then subsequently used to estimate higher values of G_i ($G_i \propto \kappa$) for EuS/Au as compared to EuS/Nb. The higher value of G_i thus obtained for EuS/Au is then used in the manuscript text to assert ‘As expected, the G_i for EuS/Au is larger than for EuS/Nb’, and hence claim validity of the arguments used to justify the experimental result.

This is in some sense - a logical fallacy.

In order to fit the experimental data, the developed theory will always predict a comparatively higher number for $\kappa_{EuS/Au}$ and therefore a higher G_i for EuS/Au; as without that the observed results cannot be fitted. This therefore cannot be used as a definitive claim for higher G_i for EuS/Au.

What would be convincing, is if G_i for the two interfaces can be measured experimentally and then these experimentally obtained numbers are used to calculate κ , which are then fed into the developed theory to see what values of $(\Delta T_c)/T_c$ values are produced as a function of Au and Nb thicknesses. If there is still a nominal match between theory and experiment, then my doubts would be quelled.

In the rebuttal, authors write the following

“Hence, $T_{c,AP}$ decreases with increasing Au thickness due to both introduction of the new Au/Nb interface with finite interfacial resistance ρ_{int} , and the inverse proximity effect. As explained in Fig. S14., without a larger interfacial exchange field at the EuS/Au interface, both ΔT_c and $\Delta T_c/T_c$, AP will be suppressed due to the introduction of the Au layer”

I am in complete agreement with the authors regarding the above, and there is no doubt from the results presented that indeed ΔT_c is not suppressed and $\Delta T_c/T_c$, AP increases. But, where is the control experiment proof regarding this? This is

extremely important, since the major claim of the paper is that the heavy metal interface is the sole important determining factor that leads to this observation.

What would prove this convincingly, is if another similar experiment is conducted with a multilayer of the type EuS/ LM / Nb/ EuS (where LM is a light metal, which can be Cu for example). If such a multi-layer does experimentally show that both ΔT_c and $\Delta T_c/T_c$ indeed decreases simultaneously, that would be very convincing proof to claim that only and only in the case when a heavy metal is used – will ΔT_c not be suppressed and hence $\Delta T_c/T_c, AP$ increases.

In summary, I am not disputing the claim that absolute spin switching is observed in this system. My doubts are solely over the experimental claims that absolute spin switching should be attributed to a heavy metal interface (since that is the title of the paper) which amplifies the exchange field. In my opinion, more control experiments and direct measurements of theoretically used parameters are required to prove causation or correlation between the two beyond reasonable doubt.

Reviewer 1

1. *“The paper reports an absolute superconducting switch effect in EuS/Au/Nb/EuS structures, where the T_c is strongly modulated by the alignment (parallel, P vs antiparallel, AP) of the magnetic moments of the EuS ferromagnetic insulating layers. The enhanced T_c modulation found in structures with an Au interlayer is discussed in terms of enhanced spin mixing conductance, namely of the exchange field, at the EuS/Au interface as compared to the EuS/Nb case. The paper contains an interesting “new” result on an “old” physical scenario enabled by the successful growth of ultrathin EuS layers in metallic multilayers. The experimental evidence shown for the absolute switch effect is compelling, and the result may be of interest for advanced quantum sensing. Before recommending acceptance, I need authors to distil the explanation.”*

We thank Reviewer 1 for noting the novelty of our results and potential implication in quantum sensing technologies.

2. *“The critical temperature of the stack is modelled with the quasi classical Green’s functions technique. Authors argue that the absolute switch effect results from an enhanced exchange field at the Au/EuS interface. The exchange field is expressed in terms of the interfacial interaction parameter $\kappa_{\text{EuS/N}}$ and the thickness of the layer $d_{\text{S/N}}$ as $h_{\text{S/N}}(d) = \kappa_{\text{int,S/N}}/d_{\text{S/N}}$. Although a slightly larger $\kappa_{\text{EuS/N}}$ is found in the case of Au interface ($\kappa_{\text{EuS/Nb}} \sim 1.2 \text{ meV}\cdot\text{nm}$ and $\kappa_{\text{EuS/Au}} \sim 1.5 \text{ meV}\cdot\text{nm}$), the rather thick (20 nm) Au layer used in the case of absolute switch, should decrease the exchange field to values well below the case of pristine EuS/Nb interface. At the same time, it is found that the T_c of the Au/Nb (4nm) stack is substantially depressed by S-N proximity effect from $T_c=5.5\text{K}$ (no Au) to $T_c = 1.86 \text{ K}$ (20 nm Au). Can authors discuss the relative importance of modified exchange field and proximity suppressed T_c in the samples with Au interlayer? Can both effect be separated (may be with an ultrathin insulating layer in between of EuS and Au)?”*

We thank the reviewer for highlighting this important point. In lines 197-216, we discuss both the modified exchange field at the EuS/Au interface and the enhanced inverse proximity effect introduced by the layer of Au. Our analysis suggests that while both effects are essential to achieving the absolute switch effect, the modified exchange field plays a more decisive role.

This conclusion is underpinned by a direct comparison of the change in ΔT_c between Fig. 2b ($\Delta T_c \sim 1.6$ K without Au) and Fig. 2d ($\Delta T_c \sim 2$ K with Au) in the main text. The increase in ΔT_c with Au cannot be explained solely by the inverse proximity effect, as this effect is independent on the magnetisation orientation of the ferromagnetic EuS layers (parallel or antiparallel). Instead, the enhancement of ΔT_c can be attributable to an increase in spin-mixing conductance at the EuS/Au interface, which modifies the exchange field acting on the superconductivity.

In Fig. S14 (copied below), we quantify this difference. Assuming that the magnetic exchange field at both Nb/EuS and Au/EuS interfaces match (1.2 meV·nm; green line), we would expect similar superconducting switch efficiencies (indicated by the intersection of the dashed line with the green line) between stacks with and without Au; however, for the structure with a 17-nm-thick layer of Au, $T_{c,AP}$ is notably reduced due to the inverse proximity effect. Theoretical analysis further suggests that with $\kappa_{\text{EuS/Nb}} = \kappa_{\text{EuS/Au}} = 1.2$ meV·nm, an absolute superconducting switch should eventually be achieved by increasing the Au layer thickness, though this would significantly lower $T_{c,AP}$ due to the inverse proximity effect. Consequently, the discrepancy between the blue ($\kappa_{\text{EuS/Au}} = 1.5$ meV·nm) and green ($\kappa_{\text{EuS/Au}} = \kappa_{\text{EuS/Nb}} = 1.2$ meV·nm) lines directly captures the additional impact of the modified exchange field at the EuS/Au interface.

We have added additional information on these points in the revised version of the paper of Supplementary Fig. S14 on lines 159-165 in S.I.

Figure S14. Superconducting switch efficiency of EuS/Au(d_{Au})/Nb(4 nm)/EuS switch with modified exchange field. For $\kappa_{\text{EuS/Au}} = 1.2$ meV·nm, the device achieves the same superconducting switch efficiency as a device without Au insertion at $d_{\text{Au}} \sim 17$ nm, though with a reduced $T_{c,AP}$. For $\kappa_{\text{EuS/Au}} = 1.5$ meV·nm, an absolute superconducting switch is achieved with $d_{\text{Au}} > 15$ nm.

3. “Another point which needs further discussion is the effect of the magnetic domain state of the EuS on the absolute switch effect. A single EuS layer in the domain state produces also substantial T_c modulation and magnetoresistance effect. This is addressed in Figure 1 g and h for the case of samples with no Au interlayer. In this regard, it is proposed that the effect is maximized when superconductor layer thickness is below the coherence length, however, the size of magnetic domains is not quoted, neither its effect is discussed.

3.1 Can authors describe the domain state of EuS? Is it modified in presence of the superconductor? How about magnetic anisotropy? What is the evidence of the canting of magnetic moments mentioned in the text?

In our previous study in Nature Communications **14**, 1630 (2023), we investigated the micromagnetic domain structure of EuS in Nb(8 nm)/EuS(30 nm) wires, grown under identical conditions, using a nanoscale SQUID on-tip (SOT) microscope. We observed a magnetic correlation length within the superconducting state that is comparable to the SOT diameter (~ 200 nm), suggesting that the EuS domains are smaller than 200 nm in diameter. As the SOT measures the out-of-plane magnetic field component, our findings indicate the presence of canted magnetic moments in EuS, likely at the surface. This observation aligns with prior reports of canted surface moments in EuS (B. Li *et al.* Phys. Rev. Lett. **115**, 067201 (2015)). Nevertheless, the overall magnetisation remains predominantly in-plane with a small out-of-plane magnetic remanence as shown in Fig. S1 (copied below).

Figure S1. In-plane (a) and out-of-plane (b) magnetic hysteresis loops of NbO_x(3 nm)/EuS(20 nm)/Nb(4 nm)/EuS(10 nm)/SiO₂//Si structure at 2.15 K. Inset of b is the same out-of-plane $M(H)$ hysteresis loop with a smaller field range.

We have added additional information on the domain structure of EuS in the revised version of the paper on lines 113-114, 131-132, and Supplementary Fig. S1 on lines 24-27 in S.I.

3.2 It would be helpful having data on magnetometry and transport of samples with a single EuS layer AND Au interlayer. On general grounds one expects that the exchange field will be

cancelled if domain size is below the superconductor coherence length. It is probably too small ($\sim 4\text{nm}$) to be the case. What is then the reason for the magnetoresistance suppression at coercivity in samples with a single EuS layer?"

In Fig. S16 (below) we have plotted volumetric magnetisation data from two single EuS layers with thicknesses of 10 nm and 20 nm, capped with a 3-nm-thick layer of Au. These sample have a four-probe resistance that exceeds 5 k Ω at room temperature. For comparison, in Fig. S17 (below), we have plotted magnetisation and transport data for an Au(3 nm)/EuS(20 nm)/Au(16 nm)/EuS(10 nm)/SiO₂//Si device.

Figure S16. Magnetic properties of **a-c** Au(3 nm)/EuS(10 nm)/SiO₂//Si and **d-f** Au(3 nm)/EuS(20 nm)/SiO₂//Si devices. **a, d** In-plane $M(H)$ measured at 1.8 K. **b, e** In-plane remanence as a function of temperature. **c, f** In-plane coercive field as a function of temperature.

Figure S17. Left: In-plane magnetic hysteresis loop of a EuS(20 nm)/Au(16 nm)/EuS(10 nm)/SiO₂//Si device. Right: In-plane $R(H)$ measurements of a EuS(20 nm)/Au(16 nm)/EuS(10 nm)/SiO₂//Si device. The device shows a typical GMR like behaviour. Both measurements were performed at 3 K.

The referee raises an interesting question (“*On general grounds...*”) regarding magnetic domain size versus coherence length in Nb. In general, in order to observe magnetoresistance through the superconducting transition the magnetic domains should be smaller than the superconducting coherence. The zero Kelvin dirty-limit coherence length is indeed relatively short in our Nb films (estimated to be 4.6 nm range from the out-of-plane critical field of 20-nm-thick Nb), but what matters here is the Ginzburg-Landau coherence across T_c which is divergent¹. Hence, the magnetic domains will suppress superconductivity in the magnetised state with T_c partially recovered at magnetic coercivity.

We have added a note on this important point in the revised version of the paper, in lines 131-132.

4. *“Minor: Please revise sentence on lines 217- 219.”*

The sentence has been revised as: “Here, we present the main results related to the experiment, with details of the model available in the **Supplementary Information**.” (in lines 224-226)

Reviewer 2

“The article discusses the detection of the absolute superconducting switch effect in hybrid magnetic/superconducting structures by the use of a heavy metal Au layer. The results are very interesting both from a purely scientific point of view and for their potential technological interest.

The manuscript is well written and presented, the graphics are clear, and the supplementary material is sufficient to back the claims.

I would recommend this article for publication with minor revisions, if the authors answer the minor concerns stated below:”

We thank Reviewer 2 for their positive appraisal of our manuscript.

1. *“Figure 1.e. The diagram shows an NbOx/EuS/Nb structure, while the caption mentions Nb/EuS/Nb. As can be read in the text, the top Nb layer is for protecting the structure, but the authors should be consistent and the structure should be the same in the Figure, caption, and main text.”*

We agree and have updated the diagram.

2. *“Also, the caption refers to the image as a “Chemistry diagram”. The authors should add that STEM was used to produce the image.”*

We agree. The caption now reads: “STEM image from a control sample of NbO_x(3 nm)/EuS(30 nm)/Nb(20 nm)/SiO₂//Si, showing the chemistry diagram with Nb (green), Eu (blue), and O (red).” in line 88-90.

3. *“The authors should include some justification as to why an Nb protection layer was used, instead of e.g. AlO_x or some other capping material.”*

Previous studies indicate that Nb forms a dense oxide layer and is self-passivating, making it a good candidate for use as a protective layer. For example, both A. Gutfreund *et al.* Nat. Commun. **14**, 1630 (2023) and Y. Gu *et al.* PRL **115**, 067201 (2015) used Nb as a capping layer. In contrast, AlO_x deposited in our e-beam evaporator was not stoichiometric, so we chose not to use AlO_x as a capping layer. Furthermore, to maintain a high-quality vacuum in the electron beam evaporator for the growth of EuS, we need to avoid growing oxides including AlO_x.

4. *“Figure 2a, the AP states do not seem entirely symmetrical, negative fields produce a wider AP state. Is this due to the canted surface magnetic moment on EuS?”*

This is a good observation. The asymmetry matches the typical magnitude of trapped flux from the superconducting coils in our system, causing a small discrepancy between the actual and set magnetic field sweep rates. As shown in Fig. 2b, the $R(H)$ of the EuS(20 nm)/Au(20 nm)/Nb(4 nm)/EuS(10 nm) structure measured at 20 mK in a separate dilution fridge (without trapped flux) does not exhibit this asymmetry. We should have noted the asymmetry and role of trapped flux in the paper. We have added a note on the asymmetry and trapped flux from the superconducting coils in the revised version of the paper, lines 156-157.

5. *“Fs hasn't been defined: For comparison, in Fig. 3 we have plotted the superconducting switch efficiency $\Delta T_c/T_c$, AP values in this study to equivalent structures in the literature involving transition metal Fs or rare-earth Fs.”*

Thank you for pointing out this omission. “F” first appears in the abstract, line 24.

6. “ T_c of the EuS/Au/Nb sample is reduced down to 2K, compared to the sample with no Au. The authors state:

‘Moreover, the addition of the heavy metal layer Au may partially suppress T_c via the inverse proximity effect, favouring the suppression of superconductivity, and hence reduce the critical field. This may add to the suppression of superconductivity in the P state.’

It seems to me this would be easily tested by the authors with a control sample Au/Nb with no EuS.”

We agree and should have included this data in the manuscript. We have performed a systematic investigation of the T_c of Au(d_{Au})/Nb(4 nm) bilayers vs Au thickness (d_{Au}), as shown in Fig. S15. We see that T_c decreases with increasing d_{Au} , demonstrating that the addition of Au suppresses T_c via the inverse proximity effect.

Figure S15. T_c of Au(d_{Au})/Nb(4 nm)/SiO₂//Si bilayers. The red dots represent the experimental measurements, and the black line corresponds to the theoretical model based on the quasiclassical Green’s function method.

Reviewer 3

“This manuscript presents a study of superconducting spin valves with Nb as the superconductor, EuS as the ferromagnetic insulator and the novel introduction of Au layer between one of the Nb/EuS interfaces. Authors argue that Au increases the spin mixing conductance and possibly enhances the proximity induced exchange fields in Nb – which results in the Nb perpetually remaining in the metallic state in the parallel orientation of both the EuS layers. Authors term the as the ‘Absolute spin valve’ as the spin valves turn superconducting only for anti-parallel orientation of the EuS layers. Authors also claim in Figure 3, that their present work represents the highest magnitude of $\Delta T_c/T_c$ ’s (defined

as Superconducting switch efficiency or SCE) ever observed and probably is close to 100%. While it is possible that Au may indeed enhance magnetic proximity coupling in Nb; I have few glaring concerns regarding these claims which make this paper unworthy of publication. These are stated below:”

1. “Claims made in Figure 3 and the numbers presented are simply not correct. Please refer to discussions of Figure 4a of reference 18. It is clear that the SCE in that device is at least 35% ($\{1.55 - 1\}/1.55$). Similar is the case for reference 20, where Figure 3d show that SCE is at least 85% ($\{2 - 0.3\}/2$). These numbers are massively understated as approximately 20% and 40% in the manuscript.”

We thank the reviewer for pointing out this important issue. For reference 18 (now reference 21), the SCE of EuS/Al/EuS has been adjusted to $(1.55 - 1)/1.55 = 35\%$. Regarding reference 20 (now reference 22), the reviewer calculated an SCE of 85% using $T_{c,P} = 0.3$ K and $T_{c,AP} = 2$ K from Fig. 3d. However, the $T_{c,P} = 0.3$ K value was measured with an external field of 80 mT, which further suppresses $T_{c,P}$. To avoid this mistake, we note that our measurements of $R(T)$ are performed in zero magnetic field. Hence, the correct SCE from reference 20 should be about 40 %, determined from the right inset of Fig. 2, where zero field $R(T)$ data are presented.

Fig. 3 in the main text has been updated below, with corrected values cited from reference 21:

2. “In both references 18 and 20, the authors achieved a similar scenario as in this manuscript – that the parallel state T_c was below the measurable lowest temperature in their respective

measurement systems. While in Ref. 18, it is stated to be 1K, in Ref. 20 it is probably 300mK. It is very much possible that in both cases, absolute spin switches were already achieved. This is especially the case in Ref. 18, where authors explicitly estimate the exchange field to be approximately 13meV, which translates to several tens of Teslas.”

Since the measurements in reference 20 (now reference 22) were conducted down to 300 mK but not in zero field, we will focus on discussing the case in reference 18 (now reference 21). The reviewer may be correct that $T_{c,P}$ was undetectable down to 1 K, consistent with our lowest measurement temperature. However, without data below 1 K, there is no evidence to confirm this. It is equally possible that $T_{c,P}$ might recover well above 20 mK. The referees’ remark is however helpful and so we have added the following text on lines 182-183 in the revised version of the paper: “We note that in Ref. 18, $T_{c,P}$ was undetectable down to 1 K but lower temperature data is not provided and so an absolute spin-valve effect cannot be concluded”. It is unfortunate the authors did not investigate $T_{c,P}$ down to mK temperatures but it should not be a ground to raise doubts on the novelty of our new observations going down to 20 mK.

Additionally, as the reviewer suggested, we applied the method from reference 18 (now reference 21) to calculate the exchange field in our device:

In reference 18 (now reference 21), the exchange field calculation for the EuS/Al/EuS superconducting switch uses the following equation:

$$T_c/T_{c,0} = 1 - 10(\Gamma S/E_F)(\xi_s/d)$$

where $T_c = 1$ K and $T_{c0} = 1.55$ K. With parameters $S = 7/2$, $E_F = 11.6$ eV, $\xi_s = 79$ nm, $d_{Al} = 3.5$ nm, they calculated $\Gamma = 16$ meV. The exchange field h_{ex} is then given by $h_{ex} = 2 \times \Gamma S \times (a/d_s)$, yielding $h_{ex} = 13$ meV.

Using the same equation, we can derive Γ and h_{ex} for our Nb/EuS device. With $T_c/T_{c,0} = 0.5$, $S = 7/2$, $E_F = 5.32$ eV, $\xi_s = 4.6$ nm for Nb, and $d_{Nb} = 4$ nm, we obtain $\Gamma = 66$ meV and $h_{ex} = 2 \times \Gamma S \times (a/d_s) = 34.65$ meV, which is about 20 times larger than the bulk superconducting energy gap of Nb. When converted using $h_{ex} = \mu_B H$, this corresponds to $H = 599$ T, which is an unrealistically high field.

Nevertheless, the calculated Γ at the Nb/EuS interface (66 meV) is approximately four times larger than Γ at the Al/EuS interface (16 meV), which, in fact, supports that achieving an absolute superconducting switch is more feasible in our device.

3. *“More importantly, I have a major concern regarding the fact that the SCE increases with increasing Au layer thickness (Figure 4b). This data can have an alternative interpretation, when Figure 4a is taken into consideration simultaneously. This can be the Metal/Superconductor proximity effect in action that is weakening superconductivity in Nb with increasing Au thickness. This is because with increasing Au thickness, T_c (AP) is seen to be monotonically decreasing, as it should in a Nb/Au bilayer. The T_c (AP) is always higher than T_c (P), because the induced exchange fields due to the top and bottom EuS layers cancel each other in the AP orientation. If one assumes that there is some asymmetry of induced exchange fields at the top and bottom interfaces; the cancellation should be more due to EuS/Au/Nb interfaces (as per the authors argument that exchange fields in such cases increase), and hence T_c (AP) of the high thickness Au layers should not decrease by more than 1K as shown in Figure 4a. Hence, in the case that this observation is just a Nb/Au proximity effect, the massive T_c (AP) lowering is readily explained, and low/unobservable T_c (P) is due to adding up of exchange fields in the P orientation on an already metal proximity weakened superconductor.”*

The reviewer’s concern focuses on the significant decrease in $T_{c,AP}$ as the thickness of the Au insertion layer increases, attributed to the inverse proximity effect. In lines 197-216, we discuss both the modified exchange field at the EuS/Au interface and the enhanced inverse proximity effect introduced by the layer of Au. Our analysis suggests that while both effects are essential to achieving the absolute switch effect, the modified exchange field plays a more decisive role.

This conclusion is underpinned by a direct comparison of the change in ΔT_c between Fig. 2b ($\Delta T_c \sim 1.6$ K without Au) and Fig. 2d ($\Delta T_c \sim 2$ K with Au) in the main text. The increase in ΔT_c with Au cannot be explained solely by the inverse proximity effect, as this effect is independent on the magnetisation orientation of the ferromagnetic EuS layers (parallel or antiparallel). Instead, the enhancement of ΔT_c can be attributable to an increase in spin-mixing conductance at the EuS/Au interface, which modifies the exchange field acting on the superconductivity.

In Fig. S14 (copied below), we quantify this difference. Assuming that the magnetic exchange field at both Nb/EuS and Au/EuS interfaces match ($1.2 \text{ meV}\cdot\text{nm}$; green line), we would expect similar superconducting switch efficiencies (indicated by the intersection of the dashed line with the green line) between stacks with and without Au; however, for the structure with a 17-nm-thick layer of Au, $T_{c,AP}$ is notably reduced due to the inverse proximity effect. Theoretical analysis further suggests

that with $\kappa_{\text{EuS/Nb}} = \kappa_{\text{EuS/Au}} = 1.2 \text{ meV}\cdot\text{nm}$, an absolute superconducting switch should eventually be achieved by increasing the Au layer thickness, though this would significantly lower $T_{c,\text{AP}}$ due to the inverse proximity effect. Consequently, the discrepancy between the blue ($\kappa_{\text{EuS/Au}} = 1.5 \text{ meV}\cdot\text{nm}$) and green ($\kappa_{\text{EuS/Au}} = \kappa_{\text{EuS/Nb}} = 1.2 \text{ meV}\cdot\text{nm}$) lines directly captures the additional impact of the modified exchange field at the EuS/Au interface.

Figure S14. Superconducting switch efficiency of EuS/Au(d_{Au})/Nb(4 nm)/EuS switch with modified exchange field. For $\kappa_{\text{EuS/Au}} = 1.2 \text{ meV}\cdot\text{nm}$, the device achieves the same superconducting switch efficiency as a device without Au insertion at $d_{\text{Au}} \sim 17 \text{ nm}$, though with a reduced $T_{c,\text{AP}}$. For $\kappa_{\text{EuS/Au}} = 1.5 \text{ meV}\cdot\text{nm}$, an absolute superconducting switch is achieved with $d_{\text{Au}} > 15 \text{ nm}$.

4. *“Considering the above scenario, one must reflect on the question that what could be an alternative simpler realisation of the absolute spin valve effect. One can imagine, that if instead of inserting the Au layer, one can just reduce the Nb thickness in EuS/Nb/EuS multilayer structure. The reduced Nb thickness would result in reduced T_c (AP) and lower thickness of Nb would result in enormously high average exchange fields in the Nb layer when the EuS layers are parallelly aligned. For some Nb thickness not too much less than 4nm, the P orientation induced exchange fields will far exceed the paramagnetic limit of thin film Nb and result in non-observation of superconductivity at all temperatures. Considering the fact, that the authors seem to have exemplary high-quality sample growth facilities, it would be a reasonable experiment to try out. In other words, the requirement of Au layer which is proposed as the key element for realisation of an absolute spin switch, may altogether be bypassed.”*

We would like to address that the Nb used in this study is already ultrathin. To the best of our knowledge, no studies have reported a superconducting spin valve effect in Nb or other metallic (s-wave) superconductors significantly thinner than 4 nm, except for aluminium (Al). Al is an

exception because its T_c can increase with increasing disorder. As shown in reference 18, Al exhibits a large spin valve effect ($\Delta T_c = 0.55$ K) at a thickness of 3.5 nm, but only a small effect ($\Delta T_c = 0.02$ K) at 5 nm.

The mechanism of T_c reduction by further thinning Nb differs fundamentally by adding a layer of Au. While Nb has a bulk T_c of 9.15 K, when the Nb layer becomes extremely thin, mechanisms such as localisation^{2,3}, disorder, surface/interface oxidation, and impurity concentration suppress T_c . In superconductor/normal-metal (S/N) bilayers, the dominant mechanism for T_c suppression is the inverse proximity effect.

Figure S13. Electrical characterisations of a NbO_x(3 nm)/EuS(20 nm)/Nb(3 nm)/EuS(10 nm)/SiO₂//Si device. Top: $R(T)$ in zero-field cooling. Middle: $R(H)$ at 4 K. Bottom: Superconducting switch performance. $R_{AP}(T)/R_N(T)$ (in green) and $\Delta R(T)/R_N(T)$ of individual $R(H)$ scans (in pink).

In our system, when the Nb thickness is reduced below 4 nm, localisation effects become noticeable, indicated by an initial resistivity increase before the superconducting transition shown in the top panel of Fig. S13. In such cases, the coherence length of highly disordered Nb could be further reduced below its thickness. As shown in the middle and bottom panels of Fig. S13, we do observe an enhanced superconducting switch in the NbO_x(3 nm)/EuS(20 nm)/Nb(3 nm)/EuS(10 nm)/SiO₂//Si device prepared under identical growth conditions, but with lower ΔT_c (~ 0.5 K) and $\Delta T_c/T_{c,AP}$ ($\sim 12\%$) compared to the NbO_x/EuS(20 nm)/Au(20 nm)/Nb(4 nm)/EuS(10 nm)/SiO₂//Si

device. Thus, simply thinning the Nb layer does not further enhance the superconducting switch efficiency. This is an important point which we should have addressed in the original version of the manuscript. We now mentioned this in the revised version of the manuscript (see lines 165-167 and Supplementary Fig. S13, lines 95-98 in S.I.).

However, with an Au layer, the increased total conduction path thickness mitigates localisation effects. This allows the proximitised superconductor to interface with both EuS layers, experiencing a substantial magnetic exchange field at the Au/EuS interface.

5. *“Taking all these factors into account, the manuscript in its current form is not fit for publication. The introduction of Au layer and its implications especially with regards to consideration of Metal/Superconductor proximity effect and associated lowering of T_c needs to be discussed thoroughly. Moreover, there can be much simpler controlled schemes for realising the same phenomena. Hence, at this moment this looks like an incremental piece of work, which needs further experimental evidence and analysis to bolster the claims made.”*

We thank the reviewer for the comprehensive feedback. In response, we have added a discussion and additional data on the role of the Au layer and its implications regarding both the inverse proximity effect and the modulation of exchange field, resulting in suppression of $T_{c,AP}$ beyond what can be explained by the inverse proximity effect alone. Moreover, we have addressed the suggestion of alternative configurations, noting that reducing Nb thickness leads to localisation effects that compromise superconductivity, as shown in our experiments with thinner Nb layers. Therefore, the Au layer is essential for achieving the observed effects. – see Supplementary Fig. S13 and S14.

We believe that these additional analyses and data satisfactorily address the reviewer’s concerns, strengthening the manuscript and providing necessary experimental evidence to support our conclusions effectively.

Minor: we received a comment from Dr. Miodrac Kulic and added his work as reference 2.

References

1. Tinkham, M. Introduction to Superconductivity. (Courier Corporation, 2004).
2. Fukuyama, H. Localization and superconductivity. *Physica B+C* **135**, 458–463 (1985).
3. Anderson, P. W. Theory of dirty superconductors. *Journal of Physics and Chemistry of Solids* **11**, 26–30 (1959).

**Reviewer 1:**

*"I have read the revised manuscript "Realisation of de Gennes' Absolute Superconducting Switch with a*
*Heavy Metal Interface" by Hisakazu Matsuki and collaborators, and authors' rebuttal to the criticism in the*
*first round of reviews.*

*Authors have satisfactorily addressed my criticism and modified the manuscript accordingly, making, in my*
*view, a more solid case of their absolute spin switch effect.*

*To my question of the relative importance of exchange field and proximity effect in suppressing T_c , authors*
*have clarified the importance of the exchange interaction parameter. A new figure S14 makes clear that for*
*increased strength of the interfacial exchange interaction thicker Au interlayers would be necessary to*
*observe the absolute spin switch effect. This clarification helps grasping the effects underpinning the*
*absolute spin switch effect.*

*To my question on the role of domain size in depressing the critical temperature, authors have made clear*
*that the important length scale is the (diverging at T_c) Ginzburg Landau coherence length. Authors have*
*also considered my questions on domain size and anisotropy and included new magnetometry data.*

*I maintain my view on the interest of the large T_c modulation in EuS/Au/Nb/EuS structures by magnetic*
*alignment of the EuS ferromagnetic layers. The result could attract readers working on superconductivity*
*but also readers from electronics and engineering as the reported effect could be of interest for quantum*
*sensing.*

*My recommendation is to publish this manuscript in its current revised form."*

We thank Reviewer 1 for their positive appraisal of our revised manuscript and for recommending publication in
*Nature Communications.*

**Reviewer 2:**

*"The authors' answers and changes to the manuscript are satisfactory and I would be happy to recommend*
*the article for publication."*

We thank Reviewer 2 for recommending publication in *Nature Communications.*

**Reviewer 3:**

*"I have gone through the new version of the manuscript, and have carefully read through all the rebuttal*
*arguments provided by the authors. Following which, my assessment is the following:*
*1) Is absolute spin switching conclusively proven?" "Yes"*

*"2) Is absolute spin switching due to a heavy metal interface theoretically possible as per theory presented*
*in supplementary section?" "Yes"*

We are very grateful to Reviewer 3 for providing helpful comments. We have replied to their comments point-by-
point below.

*“3) Most importantly: Is absolute spin switching due to a heavy metal interface experimentally proven*
*unambiguously in this manuscript?” “No. The authors do not prove conclusively that it’s the modified*
*exchange field and not the Nb/Au proximity effect that causes the above.” The reasons are stated below:*

*a) The entire experimental evidence argument almost stands on this one point which is quoted from the*
*rebuttal: “Our analysis suggests that while both effects are essential to achieving the absolute switch effect,*
*the modified exchange field plays a more decisive role. This conclusion is underpinned by a direct*
*comparison of the change in ΔT_c between Fig. 2b ($\Delta T_c \sim 1.6$ K without Au) and Fig. 2d ($\Delta T_c \sim 2$ K with Au) in*
*the main text.””*

The concern here focuses on the decrease in T_c AP as the thickness of the Au insertion layer increases, attributed
to the inverse proximity effect. In lines 197-216 of the manuscript, we discuss both the modified exchange field
at the EuS/Au interface and the enhanced inverse proximity effect introduced by adding the layer of Au. Our
analysis on $\Delta T_c/T_{c,AP}$ (the superconducting switch efficiency) vs Au thickness suggests that while both effects are
essential to achieving the absolute switch effect, the modified exchange field plays a more decisive role.

This conclusion is supported by a direct comparison of the change in $\Delta T_c/T_c$ between Fig. 2b ($\Delta T_c \sim 1.6$ K, $\Delta T_c/T_c \sim$
35% without Au) and Fig. 2d ($\Delta T_c > 1.84$ K, $\Delta T_c/T_c \sim 100\%$ with Au) in the main text. The increase in $\Delta T_c/T_c$ with
Au cannot be explained solely by the inverse proximity effect, as this effect is independent on the magnetisation
orientation of the ferromagnetic EuS layers (parallel or antiparallel). Instead, the enhancement of $\Delta T_c/T_c$ can be
attributable to an increase in spin-mixing conductance at the EuS/Au interface, which modifies the exchange field
acting on the superconductivity.

In Supplementary Fig. S14 (copied below), we quantify this difference. Assuming that the magnetic exchange field
at both Nb/EuS and Au/EuS interfaces match (1.2 meV·nm; green line), we would expect similar superconducting
switch efficiencies (indicated by the intersection of the dashed line with the green line) between stacks with and
without Au; however, for the structure with a 17-nm-thick layer of Au, $T_{c,AP}$ is notably reduced due to the inverse
proximity effect. Theoretical analysis further suggests that with $K_{EuS/Nb} = K_{EuS/Au} = 1.2$ meV·nm, an absolute
superconducting switch should eventually be achieved by increasing the Au layer thickness, though this would
significantly lower $T_{c,AP}$ due to the inverse proximity effect. Consequently, the discrepancy between the blue
($K_{EuS/Au} = 1.5$ meV·nm) and green ($K_{EuS/Au} = K_{EuS/Nb} = 1.2$ meV·nm) lines directly captures the additional impact of
the modified exchange field at the EuS/Au interface.

Hence, $T_{c,AP}$ decreases with increasing Au thickness due to both introduction of the new Au/Nb interface with
finite interfacial resistance ρ_{int} , and the inverse proximity effect. As explained in Fig. S14., without a larger
interfacial exchange field at the EuS/Au interface, both ΔT_c and $\Delta T_c/T_{c,AP}$ will be suppressed due to the
introduction of the Au layer.

We hope our explanation above clears up any misunderstanding. It is a complex issue frankly but we appreciate
the referees thoughtful comments.

**Supplementary Fig. S14: Superconducting switch efficiency of EuS/Au(d_{Au})/Nb(4 nm)/EuS switch with modified**
 **exchange field at EuS/Au interface.** For $\kappa_{EuS/Au} = 1.2$ meV·nm, the device achieves the same superconducting
 switch efficiency as a device without Au insertion at $d_{Au} \sim 17$ nm, though with a reduced $T_{c,AP}$. For $\kappa_{EuS/Au} = 1.5$
 78 meV·nm, an absolute superconducting switch is achieved with $d_{Au} > 15$ nm.

*“(a1) In Fig 2d. The T_c AP is less than 2 K (approximately around 1.75). Hence ΔT_c cannot be 2 K and is much*
 *lesser. Reasons are stated in points below.”*

Thank you for pointing out the discrepancy between the main text and the rebuttal letter. $T_{c,AP}$ for the EuS(20
 83 nm)/Au(20 nm)/Nb(4 nm)/EuS(10 nm) structure in Fig 2d is 1.86 K as written in line 173 of the manuscript and
 84 hence ΔT_c should be at least 1.84 K ($T_{c,p} < 20$ mK as it cannot be defined by mid-point (50%) criteria) - and not 2 K
 as previously stated in the rebuttal letter. Apologies for this oversight.

*“(a2) The plot in 2d is wrong, as the green line and red dots must meet each other at the 0.5 mark on either*
 *y axes (as is the case in Figures 2b, S10 and S13). Suggestion to authors - this mistake could be due to*
 *improper selection of data of temperature sensors in the dilution fridge. Typically for the low temperature*
 *ranges, dilution refrigerators will have a sensor (mostly RuO₂) that is more accurate below approximately*
 *1.4 K; and another (mostly cernox) which is more accurate above 1.5 K. It may be the case, that for the 20*
 *nm Au sample, all R vs H data (above and below 1.4 K) was recorded with RuO₂ sensor. And RvsT data was*
 *recorded with cernox. And hence this mismatch. Between 1.6 to 2 K and above, the two sensors are known*
 *to have a substantial mismatch.*

*(a3) The above mismatch also shows up in R vs T shown in Fig. 2d and R vs H in second from bottom figure*
 *of middle panel in Fig. S11. As per the R vs T, there should be no zero-resistance state in R vs H at 1.8 K. So,*
 *this is due to some error in data collection/temperature sensor selection.”*

Response to (a2) and (a3): The green line and red dots must indeed meet at the 0.5 mark on either y axes if we
 use the normalised resistance value in the AP state and ΔR of $R(H)$ scans at a fixed temperature. We have updated
 the corrected plot using R_{AP} derived from each $R(H)$ scans in the updated Fig. 2 - copied below for your
 convenience and in Supplementary Fig. S11, also copied below for convenience.

**Fig. 2: Superconducting switch performance with or without a HM interface interlayer. a**, $M(H)$ (right axis) and
 $R(H)$ (left axis) from an unpatterned NbO_x(3 nm)/EuS(20 nm)/Nb(4 nm)/EuS(10 nm)/SiO₂//Si structure (Device 1)
 at 4.2 K. Single arrows indicate the magnetic field sweep directions and double arrows represent possible
 magnetisation directions of the top and bottom EuS layers. Top left inset: schematic cross-section of the structure.
 **b**, $R_{AP}(T)/R_N(T)$ (in green) and $\Delta R(T)/R_N(T)$ (in pink) of each $R(H)$ scan. **c**, $M(H)$ at 1.8 K (right axis) and $R(H)$ at 20
 mK (left axis) of an unpatterned NbO_x(3 nm)/EuS(20 nm)/Au(20 nm)/Nb(4 nm)/EuS(10 nm)/SiO₂//Si structure
 (Device 2). Top left inset: schematic cross-section of the structure. **d**, $R_{AP}(T)/R_N(T)$ (in green) and $\Delta R(T)/R_N(T)$ (in
 pink) of each $R(H)$ scan, showing absolute switching with $\Delta T_c/T_c(AP)$ equal to 1 (approximately). Data below 1 K
 are for the same structure measured in a dilution fridge.

**Supplementary Fig. S11(a).** Extended data of normalised $R(H)$ scans of an unpatterned $\text{NbO}_x(3 \text{ nm})/\text{EuS}(20$
 $\text{ nm})/\text{Au}(20 \text{ nm})/\text{Nb}(4 \text{ nm})/\text{EuS}(10 \text{ nm})/\text{SiO}_2/\text{Si}$ structure at temperatures across T_c . Red (black) curves indicate a
 decreasing (increasing) in-plane magnetic field.

**Supplementary Fig. S11(b).** Extended data of normalised $R(H)$ scans of an unpatterned $\text{NbO}_x(3 \text{ nm})/\text{EuS}(20$
 $\text{ nm})/\text{Nb}(4 \text{ nm})/\text{EuS}(10 \text{ nm})/\text{SiO}_2//\text{Si}$ structure at temperatures across T_c . Red (black) curves indicate a decreasing
 (increasing) in-plane magnetic field.

“(a4) Fig. S15 is very important, as it shows the effect of proximity of 20 nm Au on 4 nm Nb. From this plot
the T_c of such a bilayer seems to be somewhere in the vicinity of 1.5 K. Therefore, how is it possible that
when this same bilayer is sandwiched by two magnetic EuS layers – the T_c AP (as per DeGennes, in the AP
state, the exchange field should ideally be zero) can at most be 1.5 K. In reality, the T_c AP of the sandwiched
structure is almost always observed to be less than that without magnetic sandwich T_c . So, it is possible
that when the EuS/20 nm Au/4 nm Nb/EuS sample was grown, the thickness of Nb is definitely greater than
the Nb grown for 20 nm Au/ Nb bilayer shown in Fig. S15. Therefore, it seems that the comparison in Fig.
2a and Fig. 2d is not a like to like Nb thickness comparison.”

Response to (a4) and (a5): the thicknesses of the layers were investigated by X-ray reflectivity and by measuring
the step-edge height using atomic force microscopy on control samples. The thickness of the different layers were
*in-situ* monitored during growth using Quartz Crystal Microbalance. The thickness of each layer between
growths should therefore be consistent. Although we did our best to maintain matching growth conditions
between depositions, it is inevitable that T_c of nominally identical structures grown under equivalent conditions
may vary as samples are prepared in different deposition runs e.g. $T_{c,AP}$ of EuS(20 nm)/Nb(4 nm)/EuS(10 nm) in
Fig. S10 is about 3.85 K but in Fig. 2b is about 4.6 K (two different samples grown in the same growth conditions).

We stress, however, that even if the EuS(20 nm)/Au(20 nm)/Nb(4 nm)/EuS(10 nm) sample has a Nb thickness
that exceeds 4 nm, giving a higher $T_{c,AP}$ than the Au(20 nm)/Nb(4 nm) bilayer, the fact that we observed an
absolute superconducting switch supports our argument that the superconducting spin switch efficiency is
boosted by the enhancement of the exchange field at the Au/EuS rather than the inverse proximity effect.

The magnetic proximity effect is an interfacial phenomenon that scales inversely with the thickness of the sample
(for samples thinner than the superconducting coherence length)¹⁻³. Consider a perfectly transparent Au/Nb
interface in Equation S1, it follows that the effective exchange field introduced in the bilayer is given by

$$h = \frac{\kappa_{int,S} + \kappa_{int,N}}{d_S + d_N},$$

where $\kappa_{int,S}$ and $\kappa_{int,N}$ are interfacial parameters that depend on the materials forming the interface. Therefore,
a thicker superconducting layer necessarily leads to a weaker magnetic exchange interaction h . Hence, a higher
$T_{c,AP}$ and a lower exchange interaction are detrimental to the absolute superconducting switch.

¹ Izyumov, Y. A., Proshin, Y. N. & Khusainov, M. G. Competition between superconductivity and magnetism in ferromagnet/superconductor heterostructures. *Phys.-Uspekhi* **45**, 109 (2002).

² Bergeret, F. S., Verso, A. & Volkov, A. F. Spin-polarized Josephson and quasiparticle currents in superconducting spin-filter tunnel junctions. *Phys. Rev. B* **86**, 060506 (2012).

³ Hijano, A. *et al.* Coexistence of superconductivity and spin-splitting fields in superconductor/ferromagnetic insulator bilayers of arbitrary thickness. *Phys. Rev. Res.* **3**, 023131 (2021)

*“(a5) Fig. 2b is also suggestive of broadening of T_c . Hence, if any alternative definition of ΔT_c is applied*
*(other than comparison of 50% of transition point); the ΔT_c will be much larger and can even go to a*
*maximum of 2 K. When the comparison is so tight, this point needs to be thought of very carefully before*
*attributing the observed effect more to heavy metal interface than to Au/Nb proximity effect.”*

This is a good point. We note that using mid-point (50%) of the transition is a common criteria for extracting a
nominal value for T_c . However, even if an alternative definition of ΔT_c is applied, this will not change the fact that
an absolute superconducting switch is observed due to the insertion of Au, generating an enhancement of
superconducting switch efficiency ($\Delta T_c/T_{c,AP}$).

*“(b) A concerning but relatively minor point in comparison to the above.*

*“In our system, when the Nb thickness is reduced below 4 nm, localisation effects become noticeable,*
*indicated by an initial resistivity increase before the superconducting transition shown in the top panel of*
*Fig. S13. In such cases, the coherence length of highly disordered Nb could be further reduced below its*
*thickness. As shown in the middle and bottom panels of Fig. S13, we do observe an enhanced*
*superconducting switch in the NbO_x(3 nm)/EuS(20 nm)/Nb(3 nm)/EuS(10 nm) /SiO₂//Si device prepared*
*under identical growth conditions, but with lower ΔT_c (~ 0.5 K) and $\Delta T_c/T_{c,AP}$ ($\sim 12\%$) compared to the*
*NbO_x/EuS(20 nm)/Au(20 nm)/Nb(4 nm)/EuS(10 nm)/SiO₂//Si device.”*

*• I do not agree with these arguments due to apparently contradictory data shown by the authors*
*themselves in Fig. 1 of the manuscript. Fig. 1f, 1g and 1h clearly shows that bilayers NbO_x(3 nm)//EuS(30*
*nm)/Nb(2 nm)/SiO₂//Si and / NbO_x(3 nm)/EuS(30 nm)/Nb(3 nm)/SiO₂//Si show superconductivity at 1K and*
*1.8K, and proximity effect in terms of domain wall superconductivity is clearly visible. Hence, I find it hard*
*to believe how and why a NbO_x(3 nm)/EuS(20 nm)/Nb(3 nm)/EuS(10 nm) /SiO₂//Si cannot be made properly*
*and measured.*

*• It sounds highly improbable that the effective bilayer of EuS(30 nm)/Nb(3 nm) shown in Fig 1h should*
*have a lower T_c than the effective trilayer EuS(20 nm)/Nb(3 nm)/EuS(10 nm) shown in Fig. S13. It is likely*
*that the sample whose data is shown in Fig. S13, is not a 3nm Nb sample, but one with Nb thickness is much*
*higher. Otherwise, all of the data taken together – is simply inconsistent.”*

In EuS(30 nm)/Nb(d nm) bilayer structures, the mechanism of ΔT_c is related to the interfacial exchange field seen
by the Cooper pairs in the multi-magnetic domain state vs the single magnetic domain state. In the multi-
magnetic domain state (at the coercive field), the effective exchange field across the superconducting layer is
unlikely to cancel out (to zero). Hence, a significant effective magnetic exchange field is still present, suppressing
T_c . This is consistent with Strambini *et al.* Phys. Rev. Materials **1**, 054402 (2017) which reported large area EuS/Al
tunnel junctions: they observed a significant spin splitting of the quasiparticle density of states averaged over the
tunnel junction areas in Al even in the demagnetised state of EuS, indicating a significant pair-breaking effect.
Hence, the results of a EuS/Nb bilayer and a EuS/Nb/EuS spin switch are different and should not be treated as
being equivalent.

The spin switch with a 3-nm-thick Nb interlayer indicates a localisation effect and a 4-nm-thick layer of Nb does
not. Hence, the 4-nm-thick Nb (nominal thickness) device is indeed thicker than the 3-nm-thick Nb device.

*“• As a side note. What is really the point of Fig. 1 in this paper?”*

*This figure has nothing to do with a De Gennes spin valve. The fact that exchange field exists in EuS/Nb*
*samples, is also evident from trilayers. The RH plots shown in Fig. 1 involves completely different physics of*
*domain wall superconductivity which is discussed briefly and is kind of irrelevant in the context of this*
*paper.”*

We agree that (e) - (h) in Figure 1 do not directly relate to the De Gennes spin-valve effect. The figure overall
provides basic information about the Nb/EuS structures. Some parts could be moved to the Supplementary
Information; however, we feel it is important to highlight (e) - (h) in the main paper. In our community, it has long
been assumed that superconductor/magnetic proximity effects in Nb on EuS would be weak vs Al on EuS, since
Nb is a heavier metal. We would like to keep this figure in the main paper as we believe it contains helpful
information.

*Other points:*

*“For reference 18 (now reference 21), the SCE of EuS/Al/EuS has been adjusted to $(1.55-1)/1.55 = 35\%$.*
*Regarding reference 20 (now reference 22), the reviewer calculated an SCE of 85% using $T_{c,P} = 0.3$ K and*
*$T_{c,AP} = 2$ K from Fig. 3d. However, the $T_{c,P} = 0.3$ K value was measured with an external field of 80 mT,*
*which further suppresses $T_{c,P}$. To avoid this mistake, we note that our measurements of $R(T)$ are performed*
*in zero magnetic field. Hence, the correct SCE from reference 20 should be about 40 %, determined from*
*the right inset of Fig. 2, where zero field $R(T)$ data are presented.” **“Agreed”***

*“It is unfortunate the authors did not investigate $T_{c,P}$ down to mK temperatures but it should not be a ground*
*to raise doubts on the novelty of our new observations going down to 20 mK.” **“Agreed.** The value of*
*exchange field of 13.6meV in now Ref. 21 actually is suggestive that since it is so much higher than gap*
*value of Al – even if it is measured down to dilution temperatures, it would probably be still non-*
*superconducting. Nevertheless, the reasons for not agreeing with the authors analysis is not based on what*
*has/has not been done before; but based purely on the data presented in this manuscript and its*
*supplementary information. So in light of the above observations (mainly the ones stated in point a)), I am*
*certain of data inconsistencies in this manuscript; which might have led to fallacious comparisons and*
*therefore inaccurate conclusions. This manuscript is therefore currently not fit for publication.*

We thank the reviewers’ acknowledgment of our responses to these questions in the previous round.

Response to referees letter

Reviewer #3

- (a) *The authors use their theory (over which I have no doubts) to estimate $\kappa_{\text{EuS/Nb}} \sim 1.2 \text{ meV}\cdot\text{nm}$ and $\kappa_{\text{EuS/Au}} \sim 1.5 \text{ meV}\cdot\text{nm}$ (presumably from fits to experimental data). These fitted parameter values are then subsequently used to estimate higher values of G_i ($G_i \propto \kappa$) for EuS/Au as compared to EuS/Nb. The higher value of G_i thus obtained for EuS/Au is then used in the manuscript text to assert ‘As expected, the G_i for EuS/Au is larger than for EuS/Nb’, and hence claim validity of the arguments used to justify the experimental result.*

This is in some sense - a logical fallacy.

In order to fit the experimental data, the developed theory will always predict a comparatively higher number for $\kappa_{\text{EuS/Au}}$ and therefore a higher G_i for EuS/Au; as without that the observed results cannot be fitted. This therefore cannot be used as a definitive claim for higher G_i for EuS/Au.

What would be convincing, is if G_i for the two interfaces can be measured experimentally and then these experimentally obtained numbers are used to calculate κ , which are then fed into the developed theory to see what values of $(\Delta T_c)/T_c$ values are produced as a function of Au and Nb thicknesses. If there is still a nominal match between theory and experiment, then my doubts would be quelled.

- (b) *In the rebuttal, authors write the following*

“Hence, $T_{c,AP}$ decreases with increasing Au thickness due to both introduction of the new Au/Nb interface with finite interfacial resistance ρ_{int} , and the inverse proximity effect. As explained in Fig. S14., without a larger interfacial exchange field at the EuS/Au interface, both ΔT_c and $\Delta T_c/T_c$, AP will be suppressed due to the introduction of the Au layer”

*I am in complete agreement with the authors regarding the above, and there is no doubt from the results presented that indeed ΔT_c is not suppressed and $\Delta T_c/T_c, AP$ increases. **But, where is the control experiment proof regarding this? This is extremely important, since the major claim of the paper is that the heavy metal interface is the sole important determining factor that leads to this observation.***

What would prove this convincingly, is if another similar experiment is conducted with a multilayer of the type EuS/LM / Nb/ EuS (where LM is a light metal, which can be Cu for example). If such a multi-layer does experimentally show that both ΔT_c and $\Delta T_c/T_c$ indeed decreases simultaneously, that would be very convincing proof to claim that only and only in the case when a heavy metal is used – will ΔT_c not be suppressed and hence $\Delta T_c/T_c, AP$ increases.

Response:

We thank the referee for these thoughtful observations. Please find below our combined response to points (a) and (b).

- (a) We agree that the phrasing in the original manuscript—specifically, the use of “as expected” when referring to the higher value of G_i for EuS/Au—was sub-optimal, as it may suggest a circular logic: that the theoretical fit is being used to validate the assumption that was necessary for the fit itself. To address this, we have revised the relevant sentence in the manuscript to accurately reflect our approach and avoid implying undue inference.

“For $d_{\text{Au}} \geq 15 \text{ nm}$, we are able to obtain a complete suppression of $T_{c,P}$ with $T_{c,AP}$ nonzero for an optimised induced exchange coupling with $\kappa_{\text{EuS/Au}} = 1.5 \text{ meV}\cdot\text{nm}$ and $\kappa_{\text{EuS/Nb}} = 1.2 \text{ meV}\cdot\text{nm}$, equivalent to $G_i = 2.15 \times 10^{13} \Omega^{-1}\text{m}^{-2}$ at EuS/Au and $G_i = 1.6 \times 10^{13} \Omega^{-1}\text{m}^{-2}$ at EuS/Nb interfaces⁵⁰, where G_i for EuS/Au is larger than for EuS/Nb.”

- (b) We appreciate the reviewer’s suggestion that an independent experimental measurement of G_i could be used to calculate κ values fed into the model. Unfortunately, as also noted by Referee #1, obtaining clean, well-characterised EuS/metal interfaces is challenging, and experimental measurements of G_i at these interfaces—especially for light metals—are scarce.

The reviewer further proposes a control experiment in which a lighter metal with weaker spin-orbit coupling replaces Au. While we agree that such a study could be useful, we note that previous works (e.g., Gómez-Pérez et al., *Nano Lett.* **20**, 6815–6823 (2020); Strambini et al., *Phys. Rev. Mater.* **1**, 054402 (2017)) report similar values of the induced exchange field at EuS interfaces with both light (e.g., Al) and heavy (e.g., Pt) metals, suggesting that the interface quality plays an important role in determining G_i . To our knowledge, G_i or κ values for Cu/EuS interfaces have not been reported elsewhere.

Nevertheless, during the first author's PhD we did investigate a control experiment, replacing Au with Al, a lighter element with significantly lower spin-orbit coupling than Au. We note that previous reports suggest $\kappa_{\text{Al/EuS}}$ values of approximately 0.665 meV·nm (Strambini et al., *Phys. Rev. Mater.* **1**, 054402 (2017)) and 0.7–0.9 meV·nm (Gómez-Pérez et al., *Nano Lett.* **20**, 6815–6823 (2020)). These values are smaller than the κ inferred for Nb/EuS in our study (1.2 meV·nm), and our theory (see Supplementary Fig. S14) predicts that inserting such a layer should reduce superconducting switch efficiency.

Our control experiment involved inserting an 8-nm-thick Al spacer in a EuS(20)/Al(8)/Nb(4)/EuS(10) structure (numbers in nm units). The resulting spin-valve has a suppressed ΔT_c of 0.38 K and much reduced superconducting switch efficiency $\Delta T_c/T_{c,AP}$ of 10% (with $T_{c,AP} = 3.78$ K). These results are shown in Supplementary Figure XXX and provide strong evidence that the switch efficiency is boosted by inserting a heavy metal layer. **The spin-switch efficiency for the Al control structure is also added to Figure 4(b) in the main paper.** This supports our assertion that enhanced interfacial exchange coupling—not spin-orbit coupling strength per se—is the dominant factor.

List of changes:

1. In the penultimate paragraph of the paper (lines 206– 224), we now discuss a control experiment in which we have replaced Au with the lighter metal Al. We also discuss the likely mechanism to explain why interfacial G_i , chemistry and spin orbit coupling lead to the absolute spin-valve effect:

*To further investigate the role of the interfacial layer in enabling the absolute superconducting spin-valve effect, we performed a control experiment in which the heavy metal Au was replaced with the lighter metal Al in a NbO_x(3 nm)/EuS(20 nm)/Al(8 nm)/Nb(4 nm)/EuS(10 nm)/SiO₂//Si structure. Al has negligible spin-orbit interaction (Fulde, *Advances in Physics* **22**, 667–719 (1973)) and, being lighter than Au, lacks the strong interfacial effects typically associated with heavy elements. These structures exhibited a significantly reduced switching efficiency, with $\Delta T_c/T_{c,AP}$ dropping to ~10% (see Figs. 3 and 4(b) and Supplementary Fig. S14). This result highlights the important roles of both the interfacial chemistry and interfacial exchange parameter G_i in facilitating the efficient transmission of the magnetic proximity effect from EuS into the superconducting Nb layer.*

*The mechanism underlying this proximity exchange transmission effect is related to s–d orbital exchange interaction at the F/ metal interface, where localized f-electron moments in EuS couple to the itinerant electrons in the adjacent normal metal layer. This coupling transmits the exchange field across the interface and decays with increasing normal metal thickness (*Nano Lett.* **19**, 6330 (2019)). The strength of this interaction is dependent on interface quality and the specific materials involved, requiring an intimate electronic overlap between the EuS f-electrons and the “s states” of the adjacent metal. While the magnitude of the exchange field itself does not depend directly on spin-orbit coupling—and has been reported to be similar for both Al and heavier metals such as Pt (*Nano Lett.* **20**, 6815–6823 (2020); *Phys. Rev. Mater.* **1**, 054402 (2017))—our results suggest that the enhanced G_i and interface-induced spin-mixing conductance in Au are key to achieving strong magnetic proximity effects and, ultimately, the absolute switching observed in EuS/Au/Nb/EuS structures.*

- In lines 232-235 (in the previous version of the manuscript), we agree that the phrase “as expected” —when referring to the higher value of G_i for EuS/Au—was sub-optimal, as it may suggest a circular logic: that the theoretical fit is being used to validate the assumption that was necessary for the fit itself. To address this, we have revised the relevant sentence (lines 192-195) in the manuscript to accurately reflect our approach and avoid implying undue inference.

For $d_{Au} \geq 15$ nm, we are able to obtain a complete suppression of $T_{c,P}$ with $T_{c,AP}$ nonzero for an optimised induced exchange coupling with $K_{EuS/Au} = 1.5$ meV·nm and $K_{EuS/Nb} = 1.2$ meV·nm, equivalent to $G_i = 2.15 \times 10^{13} \Omega^1 m^{-2}$ at EuS/Au and $G_i = 1.6 \times 10^{13} \Omega^1 m^{-2}$ at EuS/Nb interfaces⁵⁰, where G_i for EuS/Au is larger than for EuS/Nb.

- The superconducting switch efficiency of our control experiment involving inserting an 8-nm-thick Al spacer in a EuS(20)/Al(8)/Nb(4)/EuS(10) structure is updated in Fig. 3.

Fig. 3: Literature survey of superconducting switch efficiencies for F/S/F structures with different materials combinations including transition metal ferromagnets and f -orbital ferromagnets. PCMO is $Pr_{0.8}Ca_{0.2}MnO_3$, PCCO is $Pr_{1.85}Ce_{0.15}CuO_4$, LCMO is $La_{0.7}Ca_{0.3}MnO_3$, and YBCO is $YBa_2Cu_3O_7$.

- The superconducting switch efficiency of EuS(20)/Al(8)/Nb(4)/EuS(10) structure is added to Fig. 4.

Fig. 4: Calculated superconducting switch efficiency of EuS/Au(d_{NM})/Nb(4)/EuS structures. **a**, $T_{c,P}$ (in blue) and $T_{c,AP}$ (in green) as a function of d_{Au} . **b**, $\Delta T_c / T_{c,AP}$ as a function of d_{NM} . For optimised proximity-induced magnetic exchange fields of $\kappa_{EuS/Au} = 1.5$ meV·nm at the EuS/Au interface and $\kappa_{EuS/Nb} = 1.2$ meV·nm at the EuS/Nb interface, absolute switching is expected for $d_{Au} \geq 15$ nm (Solid line). The dashed line in **b** corresponds to $d_{Au} = 0$. **Dark grey data with $d_{Al} = 8$ nm indicates the control structure involved inserting an 8-nm-thick Al spacer in a EuS(20)/Al(8)/Nb(4)/EuS(10) structure. The superconducting switch efficiency decreases to 10 %.**

- The details of electrical characterisations of our control experiment involved inserting an 8-nm-thick Al spacer in a EuS(20)/Al(8)/Nb(4)/EuS(10) structure is updated as Supplementary Fig. S14 (Lines 105-110 in S.I.).

Supplementary Fig. 14: Electrical characterisation of NbO_x(3 nm)/EuS(20 nm)/Al(8 nm)/Nb(4 nm)/EuS(10 nm)/SiO₂/Si device. **a** $R(T)$ in zero-field cooling after magnetising the EuS at 6 K. **b**, **c** $R(H)$ at 3.4 K and 3.7 K, respectively. **d**: superconducting switch performance. $R_{AP}(T)/R_N(T)$ (in green) and $\Delta R(T)/R_N(T)$ (in pink) of individual $R(H)$ scans.